# Heuristic energy-based cyclic peptide design

**Qiyao Zhu** [ID][1], **Vikram Khipple Mulligan**[1], **Dennis Shasha** [ID][2]*

**1** Center for Computational Biology, Flatiron Institute, New York, New York, United States of America,
**2** Courant Institute of Mathematical Sciences, New York University, New York, New York, United States of America

\* shasha@cims.nyu.edu

**Data availability statement:** The source code and data used to produce the results and analyses presented in this manuscript are available from GitHub repository: https://github.com/qiyaozhu/CyclicPeptide.

## Abstract

Rational computational design is crucial to the pursuit of novel drugs and therapeutic agents. Meso-scale cyclic peptides, which consist of 7-40 amino acid residues, are of particular interest due to their conformational rigidity, binding specificity, degradation resistance, and potential cell permeability. Because there are few natural cyclic peptides, *de novo* design involving non-canonical amino acids is a potentially useful goal. Here, we develop an efficient pipeline (CyclicChamp) for cyclic peptide design. After converting the cyclic constraint into an error function, we employ a variant of simulated annealing to search for low-energy peptide backbones while maintaining peptide closure. Compared to the previous random sampling approach, which was capable of sampling conformations of cyclic peptides of up to 14 residues, our method both greatly accelerates the computation speed for sampling conformations of small macrocycles (*ca*. 7 residues), and addresses the high-dimensionality challenge that large macrocycle designs often encounter. As a result, CyclicChamp makes conformational sampling tractable for 15- to 24-residue cyclic peptides, thus permitting the design of macrocycles in this size range. Microsecond-length molecular dynamics simulations on the resulting 15, 20, and 24 amino acid cyclic designs identify designs with kinetic stability. To test their thermodynamic stability, we perform additional replica exchange molecular dynamics simulations and generate free energy surfaces. Three 15-residue designs, one 20-residue and one 24-residue design emerge as promising candidates.

## Author summary

Cyclic peptides are circular chains of amino acid residues that are promising candidates for new therapeutic drugs. Current FDA approved cyclic peptide-based drugs are mostly derived from natural sources. However, recent work has enabled the computational design of new cyclic peptide drugs. Current *de novo* computational design methods can handle sizes of 7 to 13 residues without conformational constraints. As size increases, the exponentially growing conformational space makes conformational sampling intractable. The literature's prevalent approach of random sampling finds poor

**Funding:** QZ and VKM were funded by the Simons Foundation (https://www.simonsfoundation.org/). DS was funded by the US National Science Foundation (https://www.nsf.gov/) grants 1840761, 2304758, 1934388, the National Institutes of Health (https://www.nih.gov/) grant 1R01GM121753-01A1, and NYU Wireless (https://wireless.engineering.nyu.edu/). The funders had no role in study design, data collection and analysis, decision to publish, or preparation of the manuscript.

**Competing interests:** The authors have declared that no competing interests exist.

configurations, with the result that the success rate of finding a stable design is very low. Here, we develop an efficient search algorithm by combining tailored optimization algorithms with established energy models. Our heuristic design pipeline, CyclicChamp, produces stable cyclic peptide designs of 7, 15, 20, and 24 amino acids as validated by algorithmically-independent molecular dynamics simulations. This pipeline not only expands the structural variety for future drug development, but also paves the way for potential cyclic peptide-based enzyme design.

## Introduction

Cyclic peptides are chains of fewer than 40 amino acid residues forming one or more closed loops. One common class of cyclic peptides consists of a single loop, with the N- and C-termini connected by an amide bond. In general, mid-sized cyclic peptides stand out for their superior binding affinity and selectivity compared to small molecules [1,2]. Their still modest size reduces the likelihood of provoking an immune response and enhances the ability to traverse cellular barriers compared to large protein therapeutics [3]. Further, their characteristic cyclization imposes conformational constraints, leading to structures that are more rigid compared to their linear counterparts [4]. This rigidity reduces the entropic cost associated with ordering a disordered molecule on binding to its intended target, enhancing target affinity [5]; it also prevents adoption of alternative conformations in which the peptide may bind to off-target proteins, thus enhancing specificity [6]. In addition, the connection of the N- and C-termini makes cyclic peptides more resistant to protease degradation than linear peptides. The incorporation of non-canonical or D-amino acids can further reduce immunogenicity and enhance degradation resistance [3].

Thanks to these benefits, cyclic peptide-based therapeutics have garnered significant interest over the past two decades. Currently, over 40 such drugs are used for various applications, including as antibacterial (*daptomycin*), antifungal (*caspofungin*), and immunosuppressant (*cyclosporine A*) agents. Notably, more than 80% of these drugs originate from natural sources or are their derivatives, and very few contain non-canonical or D-amino acids [7]. To expand the diversity of such drugs, it is of great benefit to design cyclic peptides *de novo*.

The current state-of-the-art protein computational design approaches are (i) machine learning (ML) driven methods, such as those that employ deep neural networks like AlphaFold, RoseTTAFold, RFdiffusion, and ProteinMPNN [8–12], and (ii) physics-based methods, such as those implemented in the Rosetta, Osprey, and Schrödinger software packages [13–15]. Recently, diffusion-based ML methods have achieved progress on cyclic peptide prediction and design. By encoding the cyclic backbone constraint into their amino acid relative position matrix, AfCycDesign uses the underlying AlphaFold model to predict and design cyclic peptides of 7–13 residues without cross-links; the authors of this method have reported experimental validation of one design for each of these sizes (in preprint [16]). High-Fold further modifies the cyclic position matrix to predict macrocycles with disulfide bonds, and published predictions for existing PDB structures of 12–39 residues obtain RMSDs of 0.4–4.5 Å [17]. RFpeptides employs the RFdiffusion model to generate cyclic backbones, and then uses the ProteinMPNN model to design sequences, providing the additional functionality of designing protein binders; however, this model does not currently support disulphides or other cross-links. The RFpeptides developers report binders of lengths 13–16 residues that have been validated experimentally, with $K_D$ values varying from 6 $nM$ to 10 $\mu M$ (in preprint [18]).

A major limitation of such ML-based approaches is that the neural network and diffusion model training dataset consisted only of canonical L-amino acids. This renders them incapable of predicting and designing cyclic peptides composed of mixed L- and D-amino acids, or of integrating non-canonical amino acids. Given the paucity of training data for heterochiral or non-canonical peptides, no existing pure ML approach is likely to prove effective at designing mixed-chirality or non-canonical cyclic peptides *de novo* [19]. Because D-amino acids are the mirror images of L-amino acids, an L-amino acid accesses only one half of the conformational space. This means that when designing an *n*-residue peptide, ML-based methods are constrained to $1/2^n$ of the potential conformational space compared to physics-based mixed chirality design. The utility of ML tools for ranking designs to prioritize experiments is also finite: while tools like AlphaFold provide confidence scores for their predictions and designs, this score can be influenced both by the quality of the design and its resemblance to the model's training data.

Further, ML models generally lack the ability to comprehensively sample the peptide's energy landscape, while physics-based approaches can. When designing a stable peptide, it is more important that the native state has a large energy gap from the other energy minima, rather than having just low energy. Hence, it is vital to have the entire energy landscape for selecting best designs for experiments.

Physics-based approaches, which could be better generalized to new chemical building-blocks and new backbone geometries never seen before, are therefore more attractive for heterochiral and non-canonical design. In Rosetta, cyclic backbone conformations are typically sampled using a generalized kinematic closure algorithm (GenKIC) [20,21]. For an *n*-residue backbone, torsion angles of *n*−3 residues are sampled randomly, biased by the conformational preferences of the residues, and torsion angles of the remaining 3 residues are solved algebraically to ensure cyclicity. The sampled cyclic backbones are relaxed using energy models that take into account atom pair interactions and torsion angle preferences [22]. Next, Rosetta employs a Monte Carlo simulated annealing algorithm to design an optimal sequence for the relaxed backbones, considering both L- and D-amino acids [23,24].

This approach has been proven effective in achieving comprehensive sampling of macrocycles ranging from 7 to 10 residues [13]. However, as the peptide size grows, the size of the conformational search space expands exponentially, greatly reducing the likelihood of identifying a stable backbone design through random sampling. Past strategies for dealing with larger peptides have included adding disulfide cross-links to further limit the accessible conformational space (used previously for cyclic peptides of 11–26 residues [13,21]), limiting conformational searches to symmetric conformations (permitting conformational sampling up to 24 residues [25]), or combining chemical cross-links with symmetric sampling and secondary structure biases (which allowed design of a cyclic 60-mer [26]). In each case, sampling of larger structures has been achieved by reducing the generality of the method, and by imposing more prior expectations of the features present in structures of interest.

In this work, we aimed to overcome the current size limit of general cyclic peptide design, reaching a size of 24 residues with mixed chirality. No additional cross-links or expectations about symmetry or secondary structure needed to be imposed. Our design pipeline, CyclicChamp, consists of (i) sampling "good" cyclic backbones, (ii) optimizing amino acid sequences to align with the backbones, and (iii) validating folded structures of the sequences. Specifically, we focused on steps (i) and (iii), and we followed Rosetta's physics-based approach for our design pursuits. Note that the two approaches by Rosetta and AlphaFold are not mutually exclusive [27], and CyclicChamp could be used for backbone conformational sampling in conjunction with any physics- or ML-based method that can carry out sequence design.

We have designed cyclic peptides of 7, 15, 20, and 24 residues. For 7 residues, our Cyclic-Champ yielded high-quality stable designs similar to those designed and experimentally validated by Hosseinzadeh *et al.* [13]. For 15–24 residues, we validated designs through the use of microsecond molecular dynamics (MD) simulations, an algorithmically independent validation approach. These MD simulations generated stable trajectories that indicated promising designs, which were further tested by replica exchange molecular dynamics (REMD) simulations. Three 15-residue, one 20-residue, and one 24-residue designs exhibited thermodynamic stability in REMD simulations, marking them as candidates for future experimental exploration.

As an extra test for our step (iii) of validating folded structures, we have performed structure predictions for 20 PDB structures of 7–24 residues without cross-links, and achieved an average of 1.2 Å RMSD. Importantly, our method is a general one that has permitted us to design large peptide folds that are not dependent on disulfide bonds or other chemical cross-links, predefined symmetry, or human-imposed secondary structure.

## Materials and methods

The overall CyclicChamp design workflow is as follows (Fig 1A):

1. We generate a pool of *n*-residue polyglycine chains, whose initial torsion angles are sampled from a permissive, flattened glycine Ramachandran distribution (which permits conformations accessible to both L- and D-amino acids to be sampled).

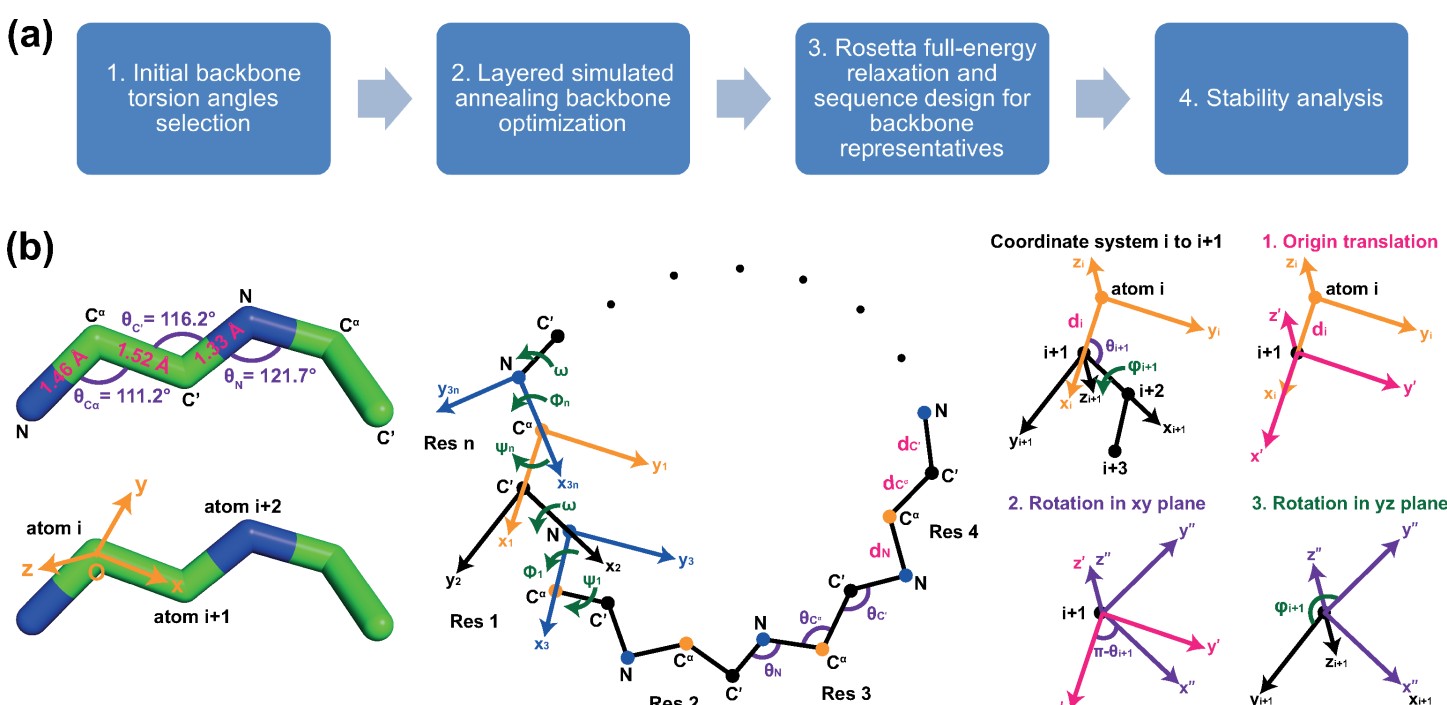

**Fig 1. CyclicChamp workflow and peptide backbone annotations.** (a) There are four steps in CyclicChamp for designing stable cyclic peptides. (b) Ideal backbone bond lengths and bond angles are assumed in CyclicChamp. For backbone closure, we consider local coordinate systems at each atom *i*. There are three steps transforming from coordinate system *i* to *i* + 1.

2. For each chain, a variant of simulated annealing is performed to search for low-energy configurations that satisfy the cyclic and hydrogen bond (H-bond) constraints.
3. We select representative low-energy configurations for relaxation and sequence design using Rosetta's FastRelax [28] and FastDesign [21,29], respectively.
4. Low-energy sequences are tested for stability by generating energy landscapes.

In the following subsections, we first derive an error function $E_{cyc}$ for the cyclic backbone constraint (Cyclic error function), and provide relevant energy functions for later backbone sampling (Backbone energy functions). Second, we describe our layered simulated annealing algorithm for low-energy cyclic backbone sampling (Efficient backbone sampling). Finally, we show the two stability analysis methods developed for cyclic peptides of different sizes (Stability analysis for small macrocycles and Stability analysis for large macrocycles).

## Cyclic error function

When tackling the backbone closure problem, we consider only the N, $C_\alpha$, and C′ atoms in each residue. A peptide backbone structure is determined by its bond lengths, bond angles, and torsion angles. Bond lengths are the most rigid [30], and are often treated as fixed values as shown in Fig 1B: bond N-$C_\alpha$ with length $d_N$ = 1.458 Å, bond $C_\alpha$-C′ with length $d_{C_\alpha}$ = 1.524 Å, and bond C′-N with length $d_{C'}$ = 1.329 Å. Bond angles and the $\omega$ torsion angle can vary to a limited extent ($\pm$5%) [30]. The ideal values are $\theta_N$ = 121.7° for the bond angle at atom N, $\theta_{C_\alpha}$ = 111.2° at atom $C_\alpha$, $\theta_{C'}$ = 116.2° at atom C′, and $\omega$ = 180°. To simplify computations, we set all these bond lengths and angles to their ideal values.

The variables are then torsion angles $\phi$ and $\psi$. To close the backbone, Go et al. showed that the torsion angles need to satisfy six independent relations [31]. Following this approach, for an $n$-residue peptide, we define $3n$ local coordinate systems corresponding to the $3n$ backbone atoms. In coordinate system $i$, the origin is set to the position of atom $i$; the $x$-axis extends towards atom $i + 1$; the $y$-axis is perpendicular to the $x$-axis and the first quadrant of the $xy$ plane contains atom $i + 2$; the $z$-axis is orthogonal to $x$- and $y$-axis using the right-hand rule (Fig 1B). The bond length from atom $i$ to $i + 1$ is denoted as $d_i$. The bond angle formed by atoms $i-1$, $i$, and $i + 1$ is denoted as $\theta_i$. The torsion angle between atoms $i-1$, $i$, $i + 1$, and $i + 2$ is denoted as $\varphi_i$.

To go from coordinate system $i$ to $i + 1$, we need an origin translation $\boldsymbol{p_i}$ of length $d_i$ from atom $i$ to $i + 1$, a counterclockwise $xy$ plane rotation by $\pi - \theta_{i+1}$, and a counterclockwise $yz$ plane rotation by torsion angle $\varphi_{i+1}$ (Fig 1B). A point with coordinate $\boldsymbol{r_{i+1}}$ in system $i + 1$ has coordinate $\boldsymbol{r_i}$ in system $i$, following

$$\boldsymbol{r_i} = \boldsymbol{T_{\theta_{i+1}}} \boldsymbol{R_{\varphi_{i+1}}} \boldsymbol{r_{i+1}} + \boldsymbol{p_i},\tag{1}$$

where

$$\boldsymbol{T_{\theta_{i+1}}} = \begin{bmatrix} cos(\pi - \theta_{i+1}) & -sin(\pi - \theta_{i+1}) & 0 \\ sin(\pi - \theta_{i+1}) & cos(\pi - \theta_{i+1}) & 0 \\ 0 & 0 & 1 \end{bmatrix},$$

$$\boldsymbol{R_{\varphi_{i+1}}} = \begin{bmatrix} 1 & 0 & 0 \\ 0 & cos(\varphi_{i+1}) & -sin(\varphi_{i+1}) \\ 0 & sin(\varphi_{i+1}) & cos(\varphi_{i+1}) \end{bmatrix}, \boldsymbol{p_i} = \begin{bmatrix} d_i \\ 0 \\ 0 \end{bmatrix}.$$

Because atom 1 is the same as atom $3n + 1$ in an $n$-residue cyclic backbone, backbone closure is then equivalent to having the coordinate systems corresponding to atom 1 and atom

$3n + 1$ be identical, *i.e.*, same origins and $x$, $y$ directional vectors (rotations preserve dot products, so $z$ is also the same). For later matrix expression, we choose system 1 to be at atom $C_\alpha$ of residue $n$ (Fig 1B), with origin being $\mathbf{0}$, $x$ directional vector $e_1 = [1, 0, 0]^T$, and $y$ directional vector $e_2 = [0, 1, 0]^T$.

The origin of system $3n + 1$ has coordinate $r_{3n+1} = \mathbf{0}$ in system $3n + 1$ and $r_1$ in system 1. To satisfy backbone closure, we require

$$r_1 = M_1 M_2 \cdots M_{n-2} M_{n-1} q + M_1 M_2 \cdots M_{n-2} q + \cdots + M_1 M_2 q + M_1 q + q = \mathbf{0}, \tag{2}$$

where $M_i = T_{\theta_{C'}} R_\omega T_{\theta_N} R_{\phi_i} T_{\theta_{C_\alpha}} R_{\psi_i}$, and

$$q = T_{\theta_{C'}} R_\omega T_{\theta_N} \begin{bmatrix} d_N \\ 0 \\ 0 \end{bmatrix} + T_{\theta_{C'}} \begin{bmatrix} d_{C'} \\ 0 \\ 0 \end{bmatrix} + \begin{bmatrix} d_{C_\alpha} \\ 0 \\ 0 \end{bmatrix}.$$

For the $x$ and $y$ directional vectors in system $3n + 1$, we require their vector forms in system 1 equal to $e_1$ and $e_2$, respectively, *i.e.*,

$$M_1 M_2 \cdots M_{n-2} M_{n-1} M_n e_1 = e_1, \quad M_1 M_2 \cdots M_{n-2} M_{n-1} M_n e_2 = e_2. \tag{3}$$

See the detailed derivation in S1 Text.

Combining requirements Eq (2) and Eq (3), and using squared error, we write the cyclic constraint into a single equation,

$$
\begin{aligned}
E_{cyc} = & \left\| q + M_1 q + M_1 M_2 q + M_1 M_2 M_3 q + \cdots + M_1 M_2 M_3 \cdots M_{n-1} q \right\|_2^2 \\
& + \left\| M_1 M_2 M_3 \cdots M_{n-1} M_n e_1 - e_1 \right\|_2^2 \\
& + \left\| M_1 M_2 M_3 \cdots M_{n-1} M_n e_2 - e_2 \right\|_2^2.
\end{aligned}
\tag{4}
$$

We call this the cyclic error function, and finding a cyclic peptide backbone solution is equivalent to finding a zero for Eq (4). Note that all the bond lengths, bond angles, and torsion angle $\omega$ are fixed to ideal values, so $M_i(\phi_i, \psi_i)$ are matrices of variables $\phi_i$ and $\psi_i$, and vector $q$ can be calculated explicitly for ideal bond angle and bond length values as $q = [3.5620, 1.3322, 0]^T$.

## Backbone energy functions

Our backbone sampling algorithm is independent of the energy model, so can work with any energy functions that are fast to evaluate. Here, we choose Rosetta's *Ref2015* energy model [22]. For the backbone atoms, we evaluate the Ramachandran, repulsive (Van der Waals), attractive (London dispersion), electrostatic, solvation, and H-bond energy terms. When analyzing backbone atom pair interactions, we consider only atom pairs separated by at least 4 covalent bonds, as is the case in the *Ref2015* energy function [22]. See detailed descriptions in S2 Text and S1 Fig.

Each amino acid has a Ramachandran map that shows the energetically allowed regions in $\psi$-$\phi$ space. Glycine has the largest Ramachandran area, proline has the smallest, and the other amino acids have similar areas but different energetically favorable regions. As L- and D-amino acids are mirror images, their accessible Ramachandran regions are mirror-symmetrical. For backbone sampling, we use the permissive, symmetrized glycine Ramachandran map to allow all possible amino acids for later sequence design (S2 Fig, regions within

the blue boundary). This map is based on the statistical distribution of glycine conformations observed in Protein Data Bank (PDB) structures but made symmetric as previously described [13,21].

## Efficient backbone sampling

**Initial torsion angles selection.** We partition the glycine Ramachandran map into six torsion bins, marking bin centers (S2 Fig). For each residue, its initial angles $\phi$ and $\psi$ are chosen randomly from one of the six centers. For an $n$ residue peptide, there are $6^n$ initial point combinations (or initial configurations). Considering equivalence classes induced by cyclic permutations (*e.g.*, "1232456" is equivalent to "2324561"), the number of unique combinations is reduced to $\frac{1}{n}\sum_{i=1}^{n} 6^{gcd(i,n)}$, where $gcd(i,n)$ is the greatest common divisor of $i$ and $n$; this is approximately $6^n/n$ for large $n$ [25,32]. For 7 residues, this yields 39,996 initial point combinations. As the macrocycle size increases, the number of combinations increases exponentially. Hence, for large macrocycles with 15, 20, or 24 residues, we randomly select 100,000 initial point combinations.

**Layered simulated annealing.** With the initial angles assigned, we search the Ramachandran space to find low-energy configurations satisfying the cyclic and H-bond constraints. To save computational cost, we devised a simulated annealing (SA) variant with multiple layers of acceptance criteria. At time step $t$, a random move within a disk of radius $k_t$ in the Ramachandran space is generated for each residue (S2 Fig). If a move enters a prohibited high-energy region of Ramachandran space (white area in S2 Fig), it is rejected for that residue. The subsequent new configuration needs to pass four layers of energy tests to be accepted: sequentially, Ramachandran energy, repulsive energy, cyclic error, and H-bond energy. For 7-residue peptides, a final energy test is added to calculate miscellaneous (attractive, electrostatics, and solvation) energies.

At each test layer $l$, we use the Metropolis criterion, *i.e.*, the new configuration passes if it has an energy $E_{new,l}$ lower than the current energy $E_l$, or below a threshold $E_{thr,l}$. If neither holds, the new configuration has a probability of $e^{(E_l - E_{new,l})/T_{t,l}}$ to pass the test, where $T_{t,l}$ is the temperature at time step $t$ and test $l$. Once a new configuration passes all tests, it is accepted and becomes the current configuration. Note that this approach is not intended to produce thermodynamic distributions of states as pure Metropolis-Hastings Monte Carlo trajectories do; instead, the goal is to rapidly discover low-energy states with the least computational expense needed, by evaluating cheaper energy terms first to reject moves.

Configurations that have low repulsive energy, low cyclic error, and strong H-bonds are recorded as good backbone candidates. Details for choosing the energy thresholds, good backbone criteria, and other simulated annealing parameters are provided in S3 Text. In particular, within the thousands of possible simulated annealing parameter combinations, we use combinatorial design [33] to select and test 51 combinations for 7- and 15-residue tests, and 400 combinations for 20- and 24-residue tests (details in S4 Text).

Once we find the optimal simulated annealing parameter combination (S1 Table), for each initial configuration, we run the layered simulated annealing algorithm. If a single run of simulated annealing does not produce any good backbone candidates, we repeat again, for a maximum of three repeats. Two example runs that successfully produced good 15-residue backbone candidates are uploaded to GitHub for illustration.

**Backbone clustering and sequence design.** There can be vast numbers of backbone candidates, and many are similar to each other. To select lowest-energy representatives, we clustered the candidates based on the torsion bins in a manner similar to that previously described [13]. Briefly, we assigned a torsion bin number to each residue in a backbone

candidate (S2 Fig), to produce a torsion bin string. For example, a string "1351246" means that the first residue in the backbone falls in torsion bin 1 of the glycine Ramachandran space, the second residue in torsion bin 3, the third residue in torsion bin 5, and so on.

We considered torsion bin strings as equivalent if they can be cyclically permuted, such as "1351246" and "3512461". To uniquely identify the equivalence class, we looked for the cyclic permutation that moves the smallest bin value to position 1. If multiple residues share the same smallest value, we chose the permutation that gives smaller value at position 2, and so on. In this way, we clustered the backbone candidates into equivalence classes of torsion bin strings. Within each class, we selected the candidate with the lowest energy as the representative.

We computed energies based on Rosetta *Ref2015*'s weights [22],

$$E_{total} = 0.45 * E_{rama} + E_{rep} + E_{hbond} + E_{other}.$$

These backbone representatives were then sent for full-energy relaxation using Rosetta's FastRelax, following past protocols [5,13,25] (scripts in S5 Text and uploaded to GitHub). Relaxed backbones with low energies were selected for further sequence design using Rosetta's FastDesign, permitting the 20 canonical amino acids (except cysteine and glycine) with both their L and D forms [13] (S6 Text). For 20- and 24-residue designs, to avoid instability caused by buried unsatisfied polar atoms, additional restrictions of amino acid types were applied [25] (S7 Text).

## Stability analysis for small macrocycles

Designed sequences having low energies underwent final stability analysis. To assess the stability of a designed sequence, we sampled alternative conformations for this sequence. The energies of these alternative conformations, together with their root-mean-square-deviations (RMSDs) from the designed structure, form the energy landscape. To calculate RMSD, we used the Kabsch algorithm [34] to align backbone heavy atoms (N, $C_\alpha$, C′, and O) of an alternative conformation with those of the designed structure.

If the lowest-energy conformations all have small RMSDs from the designed structure, then the sequence has a high chance to fold into the designed structure, and we consider the design stable. In order to provide a quantitative measure of stability, we employ the $P_{Near}$ value introduced in 2016 [21]:

$$P_{Near} = \frac{\sum_{i=1}^{N} e^{-\frac{RMSD_i^2}{\lambda^2}} e^{-\frac{E_i}{k_B T}}}{\sum_{j=1}^{N} e^{-\frac{E_j}{k_B T}}}, \tag{5}$$

where $k_B T$ = 0.62 kcal/mol (equivalent to 37 °C), $\lambda$ = 0.5 Å for small macrocycles (7 residues), $\lambda$ = 1.5 Å for medium macrocycles (15 residues), and $\lambda$ = 2.0 Å for large macrocycles (20 and 24 residues). It has been experimentally shown that $P_{Near} > 0.9$ is indicative of stability, and correlates well with experimental success in binder design [5,6,13].

For macrocycles having 7 residues, our backbone simulated annealing algorithm considered all combinations of initial angle torsion bins, so backbone sampling was comprehensive. Hence, we can expect that all low-energy conformations exist within the sampled backbones. By threading designed sequences on these backbones and using Ramachandran map at each residue as a series of filters to weed out incompatible conformations, we were able to approximate the energy landscape containing all alternative low-energy conformations.

Given that the backbones are cyclic, it was necsesary to examine all $n$ permutations of the residues for each backbone conformation. Compatible backbones were then subjected to FastRelax, following the same protocol (S5 Text), but this time with the designed sequence instead of poly-glycine. We refer to this energy landscape sampling process as *Ramachandran-stability filtering*.

## Stability analysis for large macrocycles

For macrocycles of 15–24 residues, the Ramachandran-stability filtering method failed to find a sufficient number of alternative backbone conformations due to the exponentially larger search space (see S8 Text). Simply adapting the layered simulated annealing algorithm for designed sequences was not enough. We aimed to explore energy landscapes filled with local minima, spanning both low and high RMSD regions (0–6 Å). Simulated annealing might sometimes jump across certain minima without sufficient exploration, or conversely, become trapped in some minima without investigation of others. To overcome this problem, we employed a genetic algorithm. This approach allowed us to broadly explore the landscape in the initial stages, and through successive generations, to probe and settle into the low-energy minima, thus achieving a balanced exploration of both global and local features of the energy landscapes.

The initial population of the genetic algorithm comprised alternative structures of the designed sequence, whose backbones were sampled by two separate layered simulated annealing trajectories, one targeting low energy and, when a designed backbone was available, the other targeting low RMSD. The low energy simulated annealing protocol was as described in S3 Text, while the low RMSD simulated annealing had only the cyclic error test and a RMSD test (see S8 Text). In the RMSD test, we used the Kabsch algorithm [34] to measure the backbone-heavy-atom RMSD between the new configuration and the designed structure, and used the Metropolis criterion to accept new configurations.

The sampled backbones were then subject to FastRelax, with the designed sequence specified so that the corresponding side-chains were added and optimized by FastRelax (S5 Text). We sorted the relaxed structures in ascending order of their energies and initiated energy-based clustering (see S8 Text). The $2 * N_{GA}$ lowest-energy cluster centers formed the initial genetic algorithm population.

In the genetic algorithm, each generation underwent crossover, mutation, and selection. Crossover involved checking whether a pair of parents could exchange residues within a designated region. Mutation involved random perturbation of torsion angles within a given region. See details in S8 Text and S4 Fig. After collecting the crossover and mutation children, we ran FastRelax to add side-chains and obtain full energies. Due to distortions of bond angles and lengths caused by crossover and mutation at breakpoints, we used Cartesian relaxation to restore near-ideal bond geometry (scripts provided in S9 Text). Then, we clustered the relaxed structures and selected the lowest-energy $N_{GA}$ cluster centers to form the next generation.

During each generation, of all the cluster centers, we recorded those having energies < 0 for the eventual energy landscape. We conducted 50 such generations with $N_{GA} = 500$ for 15 residue macrocycles, $N_{GA} = 750 - 5 \cdot i$ for generation $i$ of 20 residues, and $N_{GA} = 1000 - 10 \cdot i$ for generation $i$ of 24 residues. The reduced population size in later generations was for efficiency: a broad exploration of the energy landscape is beneficial for early stages, while in later stages the genetic algorithm focuses on exploring the minima, which doesn't require a large population. We refer to this energy landscape sampling algorithm as *ClusterGen*.

## Structure predictions for macrocycles in the Protein Data Bank

We modified our ClusterGen approach slightly to predict structures for existing macrocycles found in the PDB. We removed the initial simulated annealing sampling for low RMSD, which requires the designed structure as reference, so that our prediction was unbiased and the only input information was the amino acid sequence.

For large-sized PDB predictions, *i.e.*, PDB 6uf8 (12-mer), 2ns4 (14-mer), 6dzb (16-mer), and 6uf9 (24-mer), we increased the initial points for simulated annealing to 0.2, 0.3, 0.5, and 1 million, respectively, and increased the genetic algorithm population $N_{GA}$ to 1500, 2500, 5000, and 5000, respectively. This is to compensate for the removal of biased low RMSD sampling.

From each generated energy landscape, we clustered the 50 lowest-energy structures using a 1.5 Å RMSD cutoff. The five lowest-energy cluster centers were selected as the predictions. If the PDB structure is from NMR experiments and has multiple models, we selected the model that produces the highest $P_{Near}$ score for energy landscape plot.

## Molecular dynamics simulations for top designs

We next sought a means of validating designs that was independent of Rosetta or the conformational sampling methods developed here. Since currently-available ML methods for predicting macrocycle structures are incompatible with heterochiral peptides [19], we turned to molecular dynamics (MD) simulations. For 15–24 residue designs that had high $P_{Near}$ scores, we performed 1-$\mu$s MD simulations to validate kinetic stability. We used the *OpenMM v8.1.0.beta* [35] toolkit with the *amber14/protein.ff14SB* force field for peptide and the *amber14/tip3p* force field for water. The water box had periodic boundary conditions and 1 nm padding distances from the peptide. The ionic strength was set to be 0.15 molar with Na$^+$ and Cl$^-$. Full details are in S10 Text.

We also ran replica exchange molecular dynamics (REMD) simulations [36] for top designs that showed stable MD trajectories, using *OpenMMTools v0.23.1* [37] (details in S10 Text). After an initial 100 ns simulation, we plotted the radius of gyration ($R_g$) distributions for uncorrelated configurations extracted by *OpenMMTools MultiStateSamplerAnalyzer* [37] from two different time intervals 50–70 ns and 80–100 ns. If the distributions did not overlap well, we extended the REMD for an additional 50 ns.

To compute the average RMSD of a simulation, we extracted uncorrelated configurations sampled at 300 K, and calculated their $C_\alpha$-atom RMSDs from the initial design using *MDTraj v1.9.8* [38]. Free energy surfaces (FES) were derived using RMSD and $R_g$ as collective variables. We gathered RMSD and $R_g$ data from 300 K uncorrelated configurations, and computed the probability densities $P(RMSD, R_g)$ using the *histogram2d* function in the Python package *numpy* [39], with 50 bins along each dimension. The free energy was calculated as $-RT\ln\left(P(RMSD, R_g)\right)$, where $R$ is the gas constant and $T$ is the temperature (300 K). Note that when calculating the probability densities, we used the standard unit nanometer for RMSD and $R_g$, yet for visualization consistency, we plot the FES in Å.

We used the Flatiron Rusty cluster GPU nodes for the MD simulations. Each node was equipped with four NVIDIA 40 GB A100 Tensor Core GPUs (Ampere), 1024 GB system memory, and 64 CPU cores. We ran each MD or REMD simulation on one A100 GPU. Typical MD runs took about 40 hours to complete, while typical 100 ns REMD runs took about 5-9 days to complete.

## Results

Our cyclic peptide design pipeline CyclicChamp consists of several key steps: initial backbone torsion angle selection, backbone sampling through layered simulated annealing, backbone clustering using torsion bin strings, backbone relaxation with FastRelax, sequence design via FastDesign, and stability analysis by generating energy landscapes (see Fig 1). In Fig 2, we show the CPU-hours required in each step, as benchmarked on New York University's Greene High Performance Computing Cluster. A more detailed ClusterGen computation time break-down is presented in S5 Fig, and the number of candidates generated in each step is listed in S6 Fig.

The computation time for backbone sampling exhibited linear-like growth as the size of the backbone increases (Fig 2A). The time growth from 7 to 15 residues primarily resulted from a rise in the number of initial backbone configurations, escalating from 39,996 to 100,000.

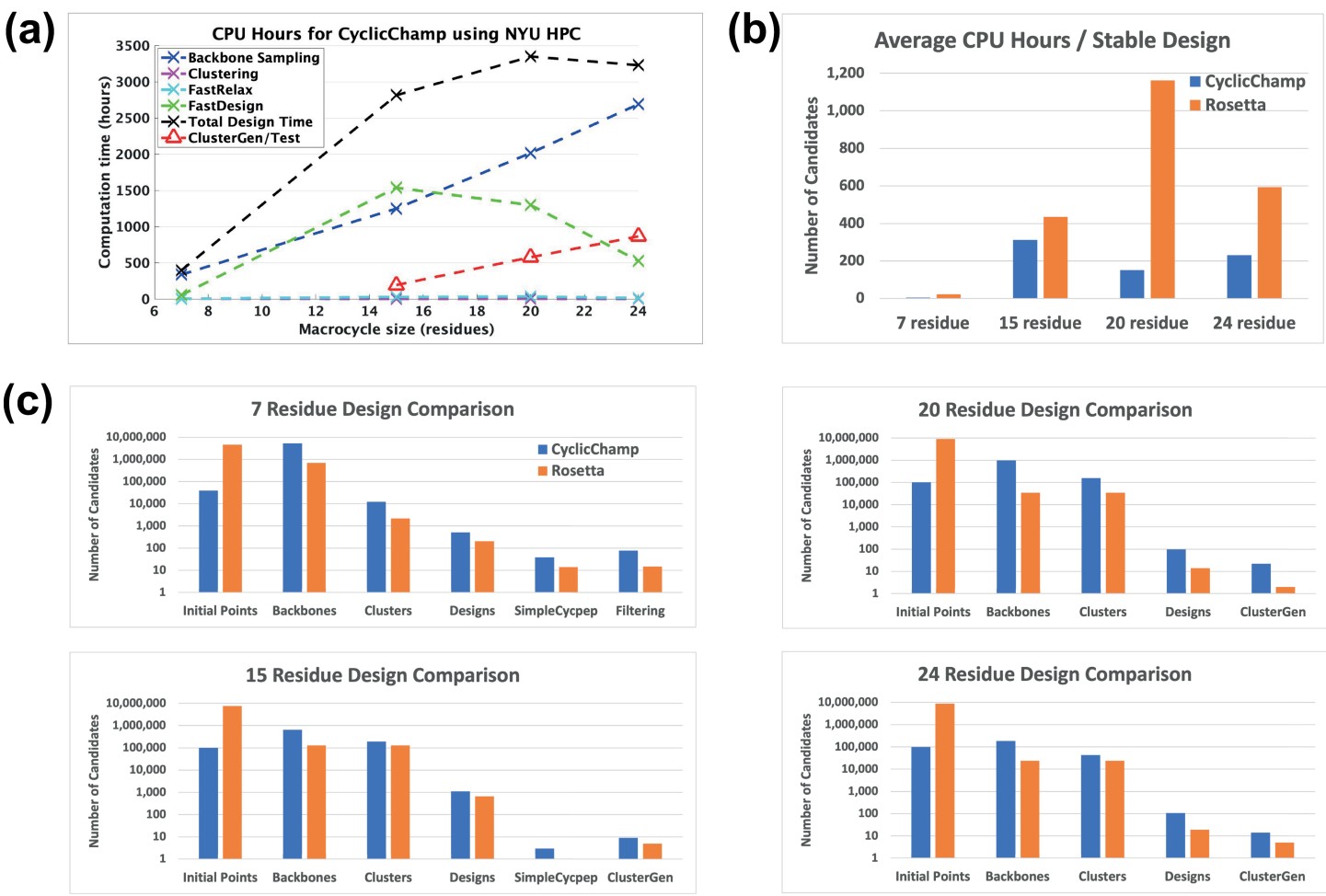

**Fig 2. CyclicChamp computation time and design comparisons with Rosetta.** (a) The computation time required by CyclicChamp backbone sampling and stability validation (ClusterGen) exhibits linear-like growth with increasing backbone size. FastDesign was faster for 20 and 24 residues than for 15 because there were fewer backbones on which we did sequence design. (b) Total design time divided by the number of stable designs validated by the filtering method for 7 residues, ClusterGen for 15 residues, and reshaped ClusterGen for 20 and 24 residues. (c) When allocating equivalent computation time for backbone sampling, CyclicChamp generated 5 to 28 times as many cyclic backbones with sufficient H-bonds as Rosetta's *simple_cycpep_predict*, which led to 2 to 11 times as many stable designs as Rosetta's after stability validation.

Beyond 15 residues, we fixed the number of initial configurations, so that the time growth was largely due to the $\mathcal{O}(N^2)$ complexity involved in calculating atom pairwise energies (see S2 Text). However, the sampling of 24-residue backbones yeilded only about one-fourth the number of backbone clusters compared to those from 15 and 20 residues, indicating a substantial increase in the difficulty of finding good backbone candidates (S6 Fig).

We conducted $P_{Near}$ stability analyses on low-energy designs using our Ramachandran-stability filtering method for 7 residues and our Clustering genetic algorithm (ClusterGen) for 15–24 residues. To obtain more stable 20- and 24-residue cyclic peptide structures, we also looked at alternative low-energy structures sampled by ClusterGen and reshaped the energy landscapes accordingly (see details in the section titled Large macrocycle 20 residue designs). Our filtering method took about 1.3 CPU-hours per stability test. The computation time of ClusterGen grew linearly as the macrocycle size increased, primarily due to the progressively larger populations adopted (see Stability analysis for large macrocycles). Rosetta's *simple_cycpep_predict* was also used as a stability test, but it failed to adequately explore the exponentially larger conformational space as the size went up to 20 and 24.

The average computation time required for obtaining a stable design with $P_{Near} > 0.9$ is plotted in Fig 2B. The reason that CyclicChamp took longer to find a stable 15-residue design than to find a 20- or 24-residue one was the more strict criteria for stability analysis, where we considered only the designed conformations, not including alternative low-energy conformations sampled by ClusterGen.

Additionally, we compared CyclicChamp's design results with Rosetta by running their design pipeline protocol described in [13]. Notably, our CyclicChamp operated more efficiently, especially for 7- and 20-residue designs, requiring only one-fourth and one-eighth the time of Rosetta's to find a stable design, respectively (Fig 2B).

The key reason for CyclicChamp's high efficiency lies in its backbone sampling (Fig 2C), which leads to significant improvements. Using the same number of CPU-hours for backbone sampling, CyclicChamp found five to twenty-eight times as many cyclic backbones with sufficient H-bonds as Rosetta. After clustering, the lowest-energy half of the cluster centers were advanced to sequence design.

CyclicChamp achieved approximately twice the number of designs for 7 and 15 residues compared to Rosetta, and approximately six times for 20 and 24 residues. After stability validation, CyclicChamp managed to produce two to eleven times the number of stable designs ($P_{Near} > 0.9$) as compared to Rosetta, illustrating its superior capability in finding high-quality cyclic peptide backbone conformations that are more likely to result in stable designs.

In the following sections, we provide stability analysis results for our 7–24 residue macrocycle designs, as well as the molecular dynamics simulation validations and the PDB structure predictions. Note that all of our designs have mixed chirality (S7 Fig). Because the sequence space accessible to heterochiral designs is exponentially larger than that accessible to homochiral, one would expect a far greater probability of the most stable "designable" backbone configurations lying in the larger heterochiral configuration space.

## Small macrocycle 7 residue designs

For the top 513 designs with computed *Ref2015* energies below –8 kcal/mol, we compared $P_{Near}$ values produced using Rosetta's *simple_cycpep_predict* application and with our Ramachandran-stability filtering method (Fig 3A). The Pearson correlation coefficient stood at $r = 0.822$, and 94% of the values exhibited deviations less than 0.4. There were 38 designs having $P_{Near} > 0.9$ by the Rosetta's method, 77 designs having $P_{Near} > 0.9$ by the Ramachandran-stability filtering method, and of these 32 had $P_{Near} > 0.9$ by both methods.

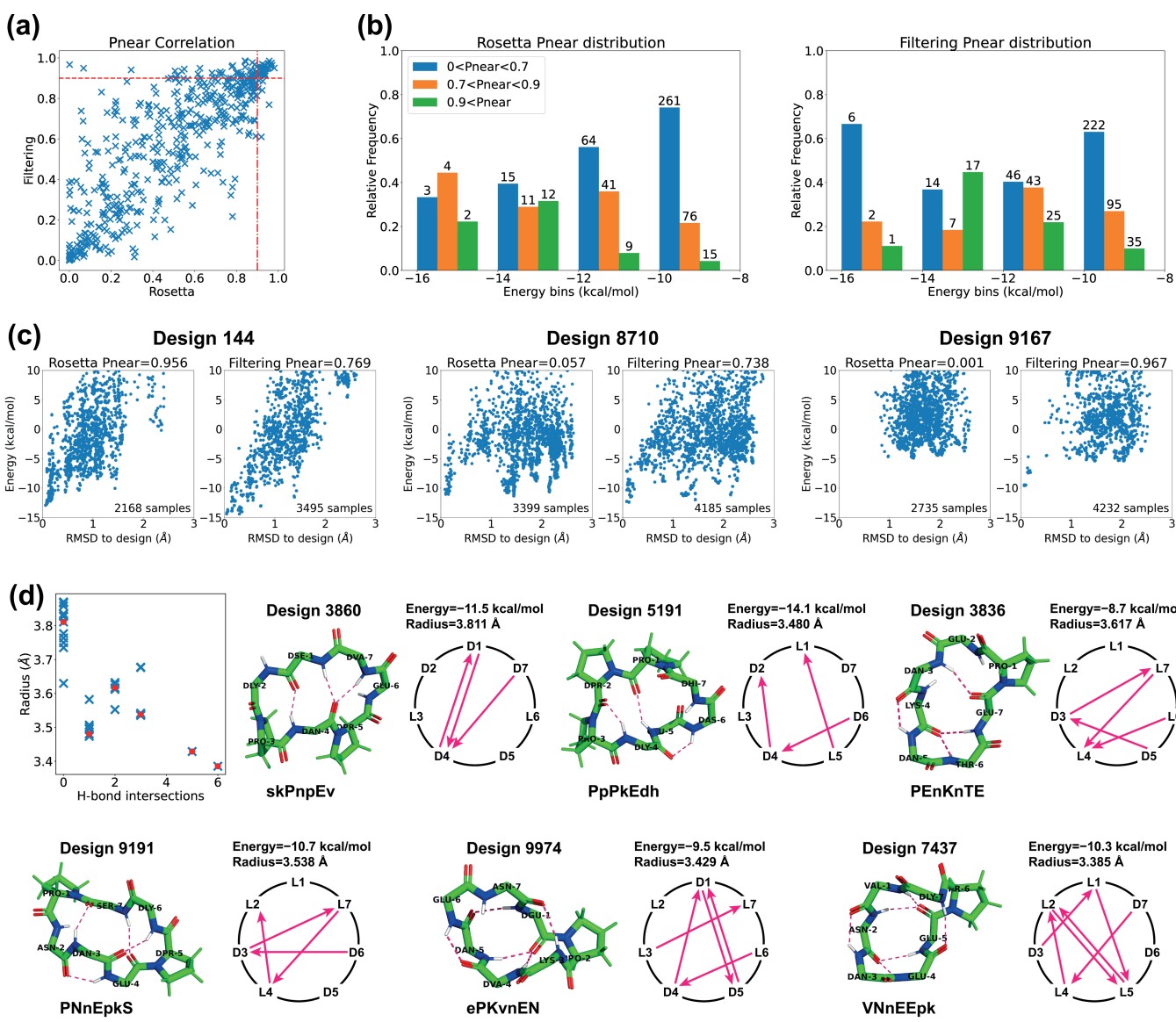

**Fig 3. Stability analysis of 7-residue designs.** (a) Correlation plot between $P_{Near}$ values calculated by Rosetta *simple_cycpep_predict* and by our filtering method. (b) $P_{Near}$ value distributions within different energy bins (kcal/mol). Design counts are labeled on top of the bars. (c) Energy landscape comparisons for three cases that have noticeable differences in $P_{Near}$ values calculated by the two methods. Left, more extensive sampling of wells further from the designed state by the filtering method results in a lower computed $P_{Near}$ value. Middle: more extensive sampling close to the designed state by the filtering method identifies a deeper minimum, raising the $P_{Near}$ value. Right: more extensive sampling by the filtering method allows exploration in a low-RMSD region missed entirely by Rosetta, identifying energy wells and raising the $P_{Near}$ value. (d) Top 7-residue designs ($P_{Near} > 0.9$) demonstrating smaller backbone root-mean-square radii with H-bond intersections. Representative designs having 0, 1, 2, 3, 5, and 6 H-bond intersections are drawn, along with their H-bond networks where L- and D-amino acids are specified and arrows point from amide proton to carbonyl oxygen. The designed sequences are written in one-letter codes, with uppercase for L-amino acids, and lowercase for D-amino acids.

We also analyzed the relationship between $P_{Near}$ values and energies (Fig 3B). We divided the 513 designs into four evenly spaced energy bins from −16 to −8 kcal/mol. Within each energy bin, we calculated the relative frequency of $P_{Near}$ values in three ranges: low ($P_{Near} < 0.7$), medium ($0.7 \leq P_{Near} < 0.9$), and high ($0.9 \leq P_{Near}$). For both methods, we found that the second-lowest energy bin possessed the largest relative probability of

high $P_{Near}$ values, suggesting that the the conformational energy of the designed state should not necessarily be as low as possible in order to achieve a design with a large energy gap between the designed state and all alternative states. This may be attributed to favourable but non-specific interactions stabilizing both the designed state and alternative states for the designs in the lowest-energy bin. We noticed that one high $P_{Near}$ design in the first energy bin dropped from 0.956 to 0.769 when switching from the Rosetta's method to the filtering method (Fig 3C, design 144). It appears that in this case the filtering method was able to sample the low-energy region more comprehensively than Rosetta could, identifying alternative low-energy structures around 0.5 Å and 1.5 Å RMSD from the design whose populations reduce the fractional occupancy very close to the designed state, lowering $P_{Near}$.

Additionally, we examined designs in which the $P_{Near}$ values of the two methods differed by more than 0.4. In most cases, the two methods yielded energy landscapes with similar shapes but different distributions (Fig 3C, design 8710). Nevertheless, there were instances in which the Ramachandran-stability filtering method alone proved capable of sampling the low-RMSD region (design 9167). While we have never seen a case in which Rosetta samples low-RMSD regions that the Ramachandran-stability filtering method misses, there is no way to check this exhaustively. Operationally, we suggest using both methods.

Overall, the Ramachandran-stability filtering method holds a distinct advantage over Rosetta's random sampling approach in scrutinizing both the low-RMSD region and all low-energy regions, and unearthing alternative low-energy structures. This is likely due to its strategy of selecting sample backbones from the candidate pool created by layered simulated annealing, characterized by their low repulsive energies and robust H-bonds. By randomly sampling the torsion angles from their Ramachandran spaces except for three residues, which are solved algebraically to ensure cyclicity through generalized kinematic closure (GenKIC) [20] Rosetta's *simple_cycpep_predict* can sometimes fail to find the low-RMSD region, especially as the dimensionality increases.

For the 32 designs that have $P_{Near} > 0.9$ by both methods, we observed an interesting correlation between the structural compactness and the H-bond patterns. We measured the structural compactness by calculating the root-mean-square distance of the backbone atoms from the backbone center of mass. When drawing the H-bond networks, we noticed that some designs have intersecting backbone H-bonds, and such designs tend to have smaller backbone radii (Fig 3D). The H-bond intersection counts varied from 0 to 6, and we show one representative design for each count. Design 7437 had the smallest radius and adopted a semi-ball shape due to its six intersecting H-bonds. Note that the compactness was not simply a product of more H-bonds. For example, despite all having three H-bonds, design 5191 with one H-bond intersection presented a radius of 3.480 Å, while design 3860 with no intersection exhibited a radius of 3.811 Å. All the 32 design structures are uploaded to GitHub, and their amino acid sequences are listed in S2 Table.

Finally, we compared our designs with the previously-published comprehensive design results from Rosetta [13] to see whether similar structures have been found. For 7 residues, that study reported 12 Rosetta-produced designs with $P_{Near} > 0.9$, of which three were experimentally validated (Design 7.1-3 in Fig 4). We aligned the backbones of our 513 designs with these 12 Rosetta designs. For each pair, we tried all seven cyclic permutations to find the best alignment. Our closest backbones have 0.114–0.639 Å RMSDs from the Rosetta design backbones, suggesting that our layered simulated annealing algorithm can find similar backbone conformations that can be stabilized with a suitable choice of sequence. Most of these simulated annealing designs had a sequence match of 2–3 residues with their corresponding Rosetta designs. This follows the observation in the 2017 Rosetta study that usually

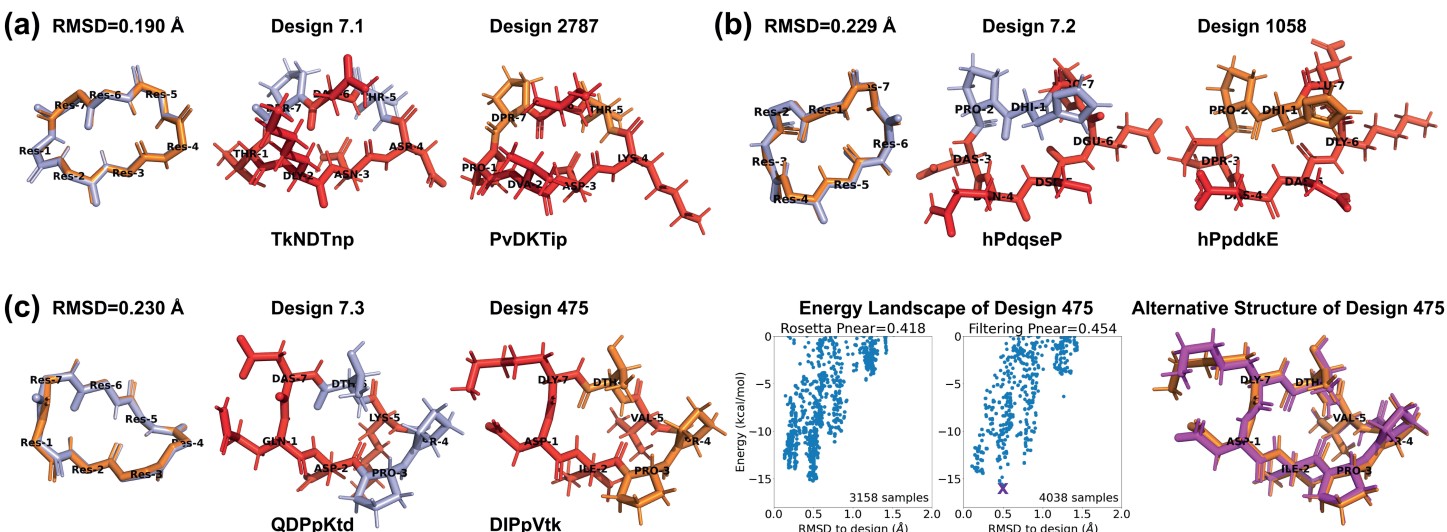

**Fig 4. Comparison with Rosetta experimentally validated 7-residue designs [13].** (a,b) From our 513 designs, we find designs (colored in orange) for which the backbones align best with the Rosetta designs (light blue). The residues having different side-chains are marked in red. (c) Design 475 has an alternate low-energy structure (purple), leading to a low $P_{Near}$ value. The amino acid sequences are written in one-letter codes, with uppercase for L-amino acids, and lowercase for D-amino acids.

fewer than three residues (often prolines) are critical for maintaining the fold [13]. Additionally, all sequences retained the same pattern of chirality (L or D-amino acids) as the Rosetta designs, with one exception that exhibited a low $P_{Near} < 0.3$ according to both Rosetta and the Ramachandran-stability filtering method. This was in stark contrast to most other designs with $P_{Near} > 0.7$, aligning with the notion that altering chirality is more disruptive than replacing an amino acid with another of the same chirality [13].

For the three experimentally validated Rosetta designs (7.1–7.3), we found closely matched simulated annealing designs with ∼0.2 Å backbone RMSDs (designs 2787, 1058, and 475 in Fig 4). At least one proline residue was preserved in each pair. Designs 2787 and 1058 maintained stable folds with $P_{Near} > 0.9$, while design 475 had only $P_{Near} \sim 0.4$. We plotted the energy landscapes for design 475, and found an alternative low-energy structure in the 0.5 Å RMSD region (Fig 4C). The main deviations between design 475 and its alternative structure were at residues 1 and 7, consistent with the observed turn flip around residue 7 in the NMR structural ensemble of Rosetta design 7.3 [13].

## Medium macrocycle 15 residue designs

Past Rosetta studies have used disulfide cross-links to design cyclic peptides of 11–14 residues [13], and structural symmetry to design larger sizes [25,26]. Our methods were able to design general stable, computationally-validated 15-residue cyclic peptides. For $P_{Near}$ stability analysis, we used both Rosetta's *simple_cycpep_predict* and our ClusterGen. In the energy landscapes, the use of Cartesian coordinate relaxation leads to an energy reduction of approximately 15 kcal/mol compared to designs using torsion angle relaxation (Fig 5B and 5D). Because of the high computational cost of validation (100,000 samples for *simple_cycpep_predict*, and 50 generations of a population of 500 for ClusterGen), we validated only the top 75 designs having the lowest energies.

A correlation plot in Fig 5A compares the $P_{Near}$ values computed by ClusterGen and Rosetta. The $P_{Near}$ values from ClusterGen were evenly distributed across all ranges, and there

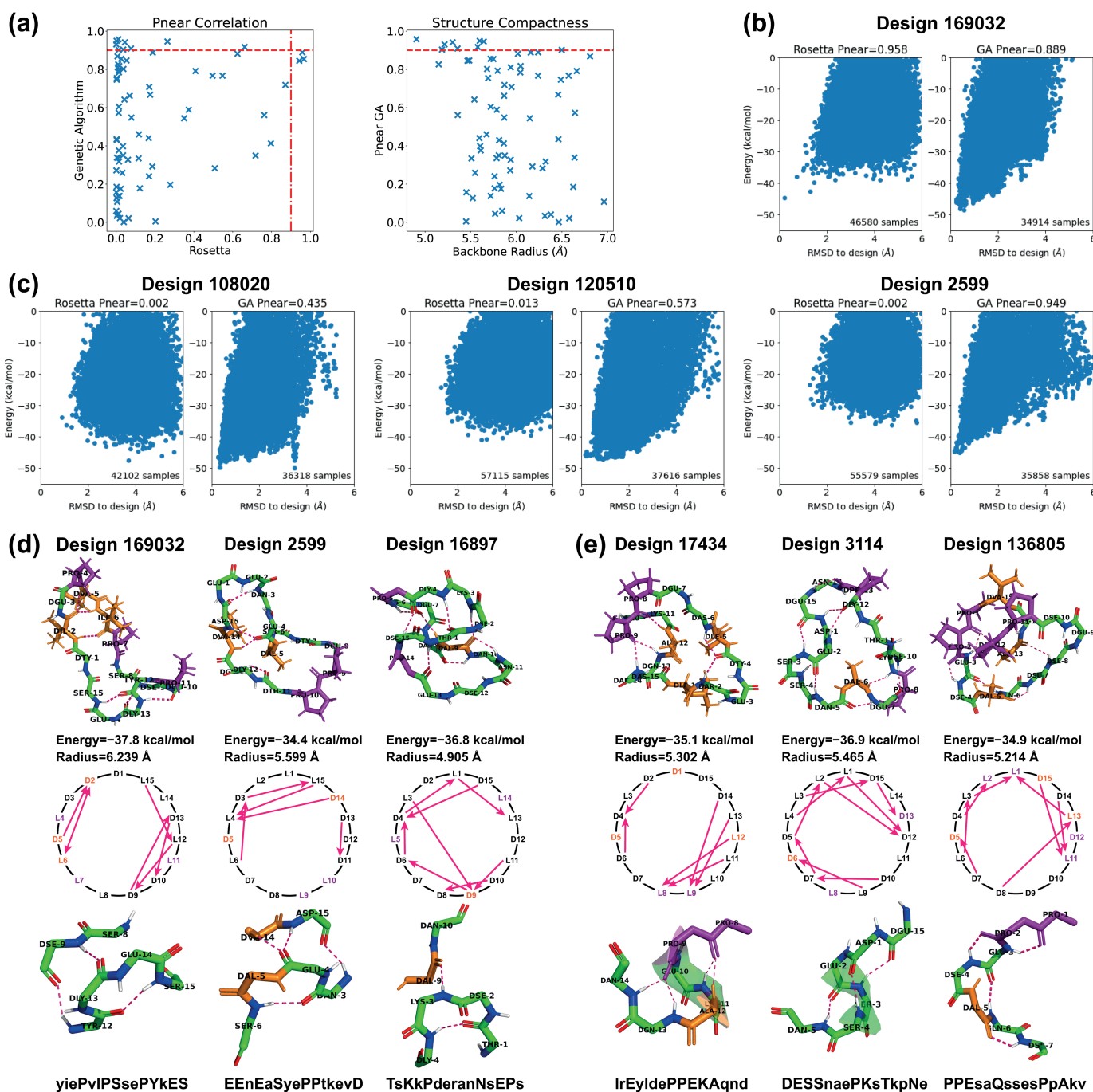

**Fig 5. Stability analysis of 15-residue designs.** (a) Correlation plot between $P_{Near}$ values calculated by Rosetta *simple_cycpep_predict* and by our ClusterGen (left). The $P_{Near}$ values of our ClusterGen are also plotted against the backbone root-mean-square radii (right). (b) Energy landscape comparison for one case in which both methods obtain high $P_{Near}$ values. (c) Energy landscape comparisons for three cases in which the two methods calculate significantly different $P_{Near}$ values. (d) Top designs with bending backbones. The backbone atoms, prolines (in color purple), and hydrophobic amino acids (ALA, ILE, LEU, VAL, MET, PHE in color orange) are shown. The backbone turn segments are enlarged. (e) Top designs with short alpha helices and consecutive $i, i + 2/i + 3$ H-bonds. The designed sequences are written in one-letter codes, with uppercase for L-amino acids, and lowercase for D-amino acids.

were nine designs with $P_{Near} > 0.9$. Meanwhile, 64% of Rosetta's $P_{Near}$ values fell below 0.1, and only three designs achieved $P_{Near} > 0.9$. These three designs also exhibited high $P_{Near}$ values according to ClusterGen, and their energy landscapes' low-RMSD regions were better explored by ClusterGen, as seen in Fig 5B. In instances where significant differences existed between the two methods' $P_{Near}$ values, Rosetta tended to underestimate $P_{Near}$ due to failure to extensively sample low-RMSD regions (Fig 5C). In contrast, ClusterGen consistently sampled the low-RMSD region, and distinguished peptides by generating energy landscapes of various shapes, such as the dual-minimum landscape observed in Design 108020, the broad energy minimum in Design 120510, and the sharp funnel shape in Design 2599. When we plotted ClusterGen's $P_{Near}$ values against the structure compactness measured by backbone radii of designed states (Fig 5A), we found that designs with small radii, particularly those under 5.5 Å, tended to exhibit high $P_{Near}$ values.

Interesting backbone structural motifs appeared in the top designs ($P_{Near} > 0.9$ by either Rosetta or ClusterGen, structures uploaded to GitHub and sequences listed in S2 Table). Designs 169032, 2599, and 16897 shared a recurring structural motif in which the backbone bends (Fig 5D). At the bending locations, we saw $i, i + 3$ H-bonds, with the CO group in residue $i$ binding to the NH group in residue $i + 3$. Such $i, i + 3$ H-bonds can cause backbone turns, as observed in the 2017 Rosetta study [13]. In these three designs, the $i, i + 3$ turns were paired with extra long-range H-bonds. The CO group of residue $i + 1$ or the NH group of residue $i + 2$ bound to the opposite backbone side, leading to a simultaneous bend on both sides. Despite the fact that all three designs have intersecting H-bonds, the H-bonds in design 16897 are closely intertwined, while the H-bonds in design 169032 can be distinctly separated into two parts. Consequently, the backbone root-mean-square radius varies from 4.905 to 6.239 Å.

Another three designs contained ordered consecutive H-bonds (Fig 5E). In design 17434, residues 13 and 14 bound to residue 9, and residues 11 and 12 bound to residue 8. These H-bonds held the twisted backbone tightly to form a short $\alpha$-helix. In design 3114, three consecutive $i, i + 3/i + 4$ turns among residues 15 and 1–5 shaped another short $\alpha$-helix. When the $i, i + 3$ turn paired with $i, i + 2$ H-bonds, there resulted a semi-circular segment as seen in residues 1–7 of design 136805. These H-bond arrangements may prove valuable as building blocks for future design endeavors.

## Large macrocycle 20 residue designs

We first validated 22 low-energy designs using both Rosetta and ClusterGen to compare the performance of the two algorithms. The $P_{Near}$ value correlation plot is shown in Fig 6A. Notably, all of Rosetta's $P_{Near}$ values fell below 0.2, while ClusterGen's $P_{Near}$ values spanned evenly from 0.001 to 0.822. This was caused by Rosetta's failure to sample the low-RMSD regions, as can be seen in the round-shaped energy landscapes in Fig 6B. We then validated 47 more low-energy designs using only ClusterGen. By plotting the ClusterGen's $P_{Near}$ values against backbone root-mean-square radius, we again saw that designs with high $P_{Near}$ values tended to have small radii (Fig 6A). The highest $P_{Near}$ value obtained was 0.897 with a radius of 6.196 Å.

As our primary goal was to find stable cyclic peptide structures, rather than to design fixed backbone conformations, we also looked at alternative low-energy structures in the energy landscapes of our designs. We reshaped these landscapes by selecting the lowest energy structure sampled as the native state for each designed sequence, and recomputed the backbone RMSD values of all samples from this new native state (Fig 6B). We then calculated the $P_{Near}$ values for these reshaped energy landscapes. Note that ClusterGen is not biased towards the

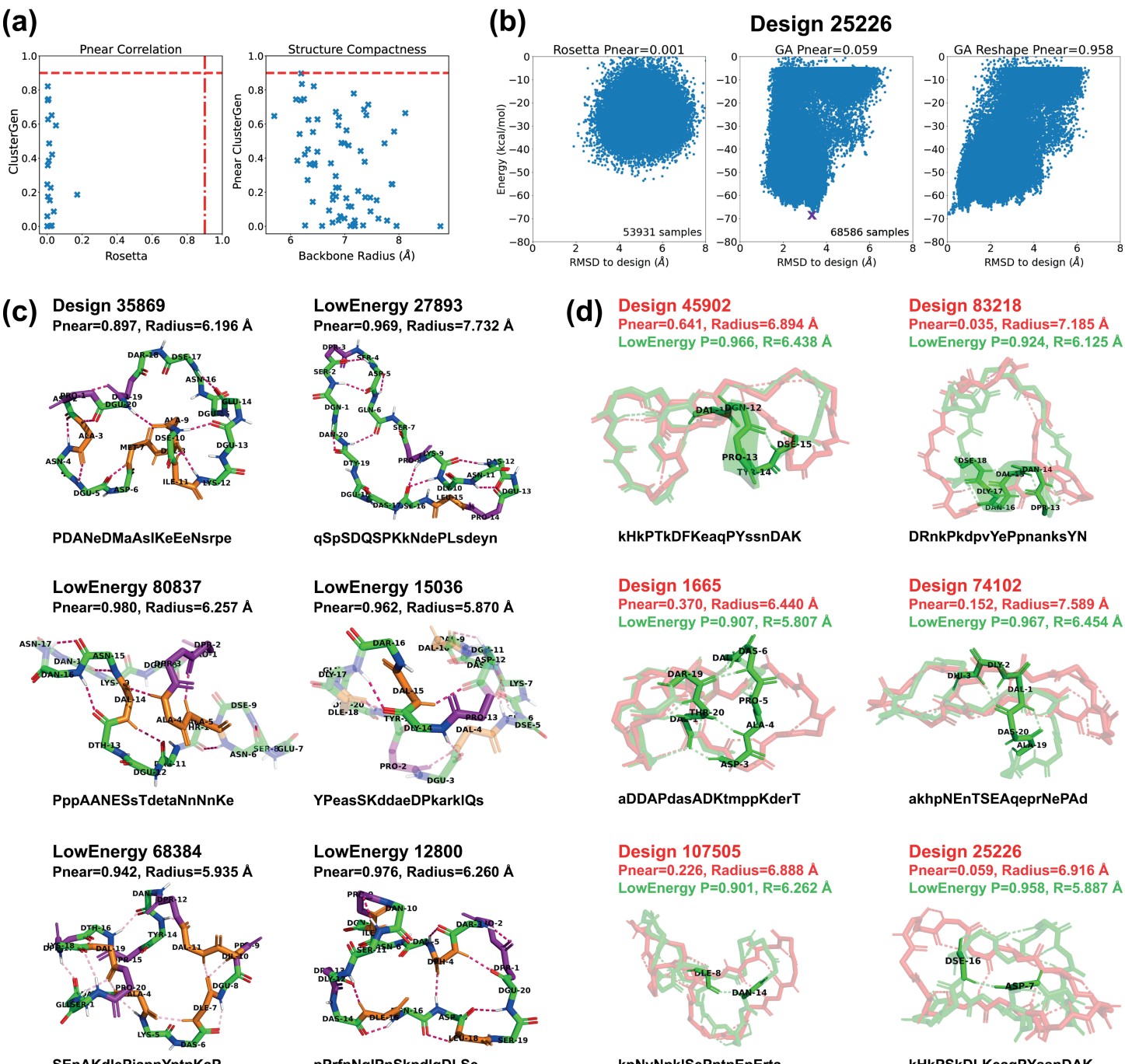

**Fig 6. Stability analysis of 20-residue designs.** (a) Correlation plot between $P_{Near}$ values calculated by Rosetta and our ClusterGen (left). The ClusterGen's $P_{Near}$ values are plotted against the backbone root-mean-square radii (right). (b) Example energy landscape comparison. By selecting the lowest-energy structures (marked by a purple cross) as the native states, the ClusterGen landscapes were reshaped. (c) Six top designs with minor conformation changes between their initial target states and the lowest-energy structures. (d) Six low-energy structures (colored in green) that show major backbone conformation changes from their designed structures (red). These involve formation of a short helix or a compact bending. The amino acid sequences are written in one-letter codes, with uppercase for L-amino acids, and lowercase for D-amino acids.

initial designed structure, because in each generation, we choose the lowest energy samples to form the next generation. As long as the original energy landscape is thoroughly explored, the reshaped landscape will reflect the stability of the new native state. This reshaping yielded 22 low-energy structures with $P_{Near} > 0.9$. From these, we selected the 11 with energies below −65 kcal/mol, as well as the top design 35869 that has an original $P_{Near} = 0.897$ before reshaping; these are presented in Fig 6C and 6D. The 22 high $P_{Near}$ structures and the design 35869 are uploaded to GitHub, and their amino acid sequences are listed in S2 Table.

Design 35869 had a hydrophobic core (residues 7–11) that formed four backbone H-bonds with the surrounding residues, imparting rigidity. Among the selected low-energy structures, five exhibited local conformational deviations from their initially-designed configurations, so we depict only their low-energy conformations in Fig 6C. Notably, structure 80837 demonstrated a quasi-cyclic (C2) symmetry within its backbone, an intriguing feature considering the sequence's inherent asymmetry. This structure formed three $i, i + 3$ backbone turns across residues 11–16 and 3–5, with an H-bond between residues 4 and 14 consolidating a hydrophobic core. Structure 68384 had the most backbone turns, five $i, i + 3$ H-bonds highlighted in Fig 6C, resulting in a twisted structure with a small backbone radius. In structure 15036, a $3_{10}$-helix was present in residues 13–16, accompanied by two $i, i + 3$ turns.

The remaining six low-energy structures displayed considerable conformational shifts from their originally designed configurations, as illustrated in Fig 6D. In structure 45902, a half-turn helix emerged, enhancing the stability of the loose backbone end. Structure 83218 underwent a transformation that introduced a complete helical turn, significantly decreasing the backbone radius from 7.185 Å to 6.125 Å, with a corresponding $P_{Near}$ value increase from 0.035 to 0.924. The other four structures formed dense H-bonding that lead to compactness. Specifically, the consecutive H-bonds between residues 3–7 and 19–20 in structure 1665 pulled the backbone segments closer. In structure 74102, an H-bond network among residues 1 to 3 facilitated a 90-degree turn, effectively altering the backbone from a flat to a more globular shape. Structures 107505 and 25226 each introduced a single long-range H-bond between residues 8 and 14, and residues 7 and 16, respectively, drawing the loose backbone ends together.

These top structures all had nicely packed hydrophobic cores made of 1–5 residues (S8 Fig). The number of proline residues varied from 2 to 5, and they were scattered around the peptide surfaces to enhance structural rigidity. Low-energy structure 68384 had the highest count of both hydrophobic and proline residues, harboring five of each.

## Large macrocycle 24 residue designs

We validated the top 11 designs using both Rosetta and our ClusterGen. Rosetta's failure to thoroughly explore the energy landscapes resulted in near-zero $P_{Near}$ values (Fig 7A). We then validated 71 more designs using ClusterGen, and the highest $P_{Near}$ value obtained is 0.896 from design 31759, with a backbone radius of 7.408 Å. Designs 19384 and 21698 also have high $P_{Near}$ values of 0.790 and 0.786. To expand our search for stable structures, we reshaped the energy landscapes by selecting the lowest-energy samples as the native states and recomputing RMSDs (*e.g.*, Fig 7B). There were 14 low-energy structures achieving $P_{Near} > 0.9$, of which 10 had energies below −75 kcal/mol, including the one for design 31759. We show these 10 structures, as well as the low-energy structures 19384 and 21698 in Fig 7C and 7D. The 14 high $P_{Near}$ structures and the low-energy 19384 and 21698 are uploaded to GitHub, and their sequences are listed in S2 Table.

Similar to the 20-residue designs, half of these 24-residue designs had only local conformational changes in their low-energy structures (Fig 7C). Structure 21698 showed a diverse array

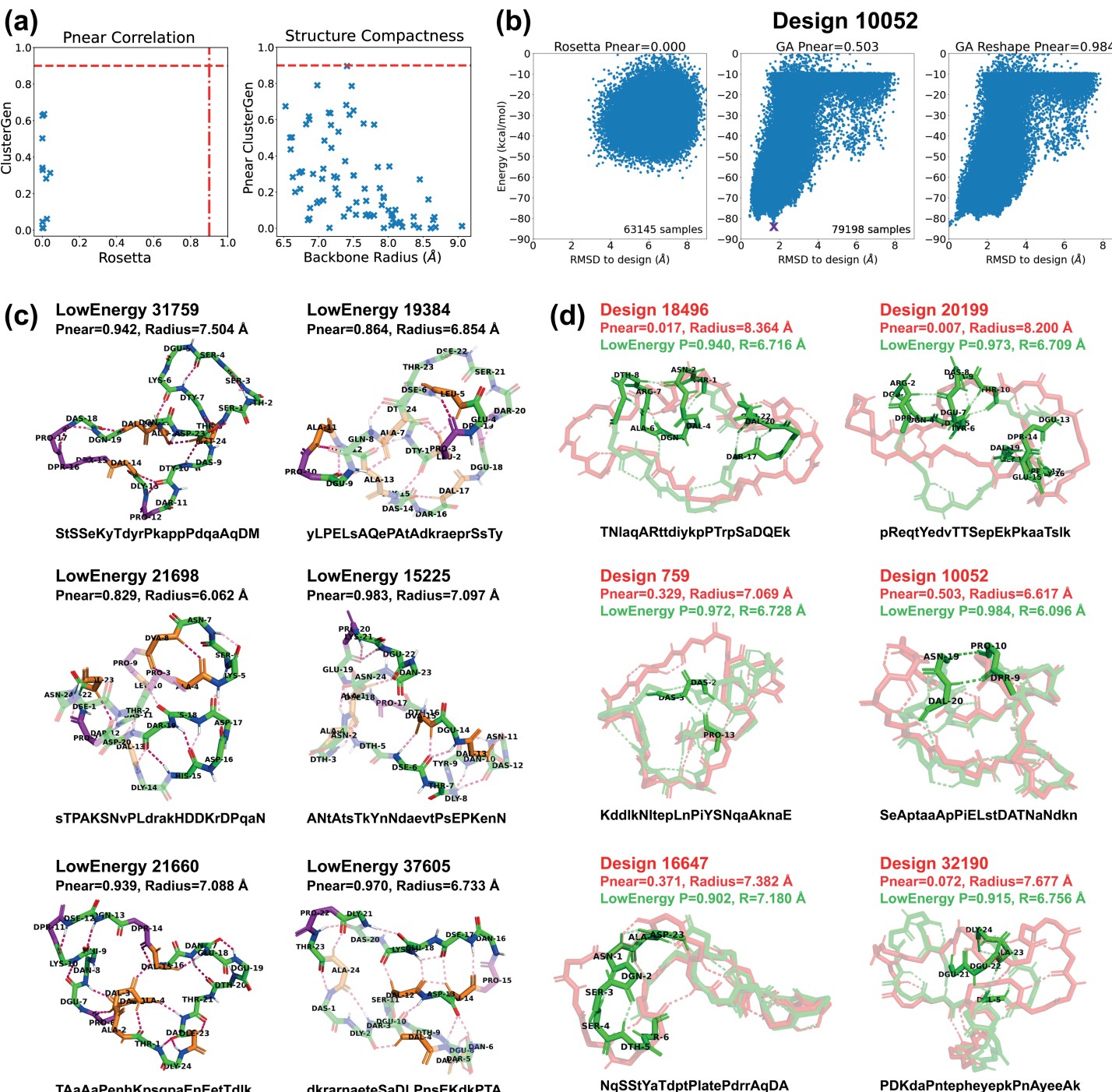

**Fig 7. Stability analysis of 24-residue designs.** (a) Correlation plot between $P_{Near}$ values calculated by Rosetta and ClusterGen (left). The ClusterGen's $P_{Near}$ values are plotted against the backbone root-mean-square radii (right). (b) Example energy landscape comparison. By selecting the lowest energy structures (marked by a purple cross) as the native states, the ClusterGen landscapes were reshaped. (c) Six top designs with minor conformation changes in their low energy structures. (d) Six low energy structures (colored in green) that show major backbone conformation changes from their designed structures (red). The amino acid sequences are written in one-letter codes, with uppercase for L-amino acids, and lowercase for D-amino acids.

of secondary structures: a $3_{10}$-helix in residues 21–23; an isolated $\beta$-bridge between residues 4 and 8 with an intermediate backbone turn across residues 5–7; and another $\beta$-bridge between residues 15 and 19, enclosing a central backbone turn. The two backbone turn regions were parallel, connected by two H-bonds, forming a layered configuration. Structure 19384 featured two $3_{10}$-helices in residues 3–5 and 9–11. In structure 15225, a $\beta$-ladder linked residues 6–7 and 13–15, and consecutive backbone turns formed in residues 20–23. Structure 37605 stood out by forming the highest number of backbone H-bonds (17 in total). Moreover, it had a $\beta$-ladder between residues 12–14 and 17–19, combined with an isolated $\beta$-bridge between residues 4 and 13. Also, a $3_{10}$-helix formed in residues 21–23.

For designs undergoing major conformational shifts from their designed configurations to their low-energy configurations, there were notable decreases in their backbone radii (Fig 7D). The largest radius reductions, approximately 1.5 Å, occured in structures 18496 and 20199. These two low-energy structures adopted shapes resembling a "palm" with three "fingers". The left two "fingers" are held close to each other by an inter-"finger" H-bond between residues 2 and 8 in structure 18496, and residues 2 and 7 in structure 20199. Other designs exhibited more modest decreases in their backbone radii, less than 1 Å. In structures 759, 10052, and 32190, the formation of long-range H-bonds played a crucial role in bridging distant segments, enhancing the structural coherence and stability. Structure 16647 had a $3_{10}$-helix forming in the less structured region of residues 2–4.

The top 24-residue structures had a greater proportion of hydrophobic amino acids than the 20-residue ones, featuring between 3 to 8 hydrophobic residues (S9 Fig): in this size range, folds with true hydrophobic cores like those of natural proteins begin to emerge. The number of proline residues remained at the same level of 2–5.

## Molecular dynamics results for top designs

We conducted 1-$\mu$s molecular dynamics (MD) simulations on the top designs of 15 residues (Fig 5), 20 residues (Fig 6), and 24 residues (Fig 7), with a timestep of 2 fs. The backbone $C_\alpha$-atom RMSD was measured every 10 ps, comparing the trajectory frame to the initial designed structure. Among the 30 trajectories analyzed, 9 exhibited relatively low RMSDs, indicating kinetic stability. These 9 RMSD trajectories are displayed in Figs 8A and S10, and the rest are shown in S11 Fig.

The 15-residue design 169032 displayed the most stable trajectory, maintaining an RMSD below 2 Å for the majority of the simulation. Its highest RMSD was 3.51 Å, yet it retained a globally similar shape to the original design. Trajectory 17434 has its RMSD rise to ~3 Å after 400 ns, adopting an enlongated conformation. Trajectory 136805 showed oscillating RMSD values between 0.27 and 5.24 Å, with frequent shifts to a slender conformation marked by a bent backbone.

For 20 residues, RMSDs generally fluctuated around the 2–4 Å level. Trajectory 27893 had the smallest fluctuation, and constantly shifted between an open and a compact shape. Trajectory 68384 started to fluctuate after 400 ns. Its low-RMSD states preserved structures closely resembling the original designs, while high-RMSD states had more expanded conformations.

For 24 residues, RMSDs stayed around 4 Å. Trajectory 21698 showed large RMSD oscillations within the first 200 ns. Trajectory 19384, similarly, underwent early RMSD fluctuations but levelled off at approximately 4 Å for the rest of the simulation. Trajectory 20199 had small oscillations for most of the time except a spike in RMSD at ~900 ns.

Overall, designs 169032 and 27893 exhibited the most kinetically stable trajectories, showing minor RMSD fluctuations. The remaining designs experienced larger RMSD variations, yet they frequently reverted to the designed structures. To explore the energy landscapes

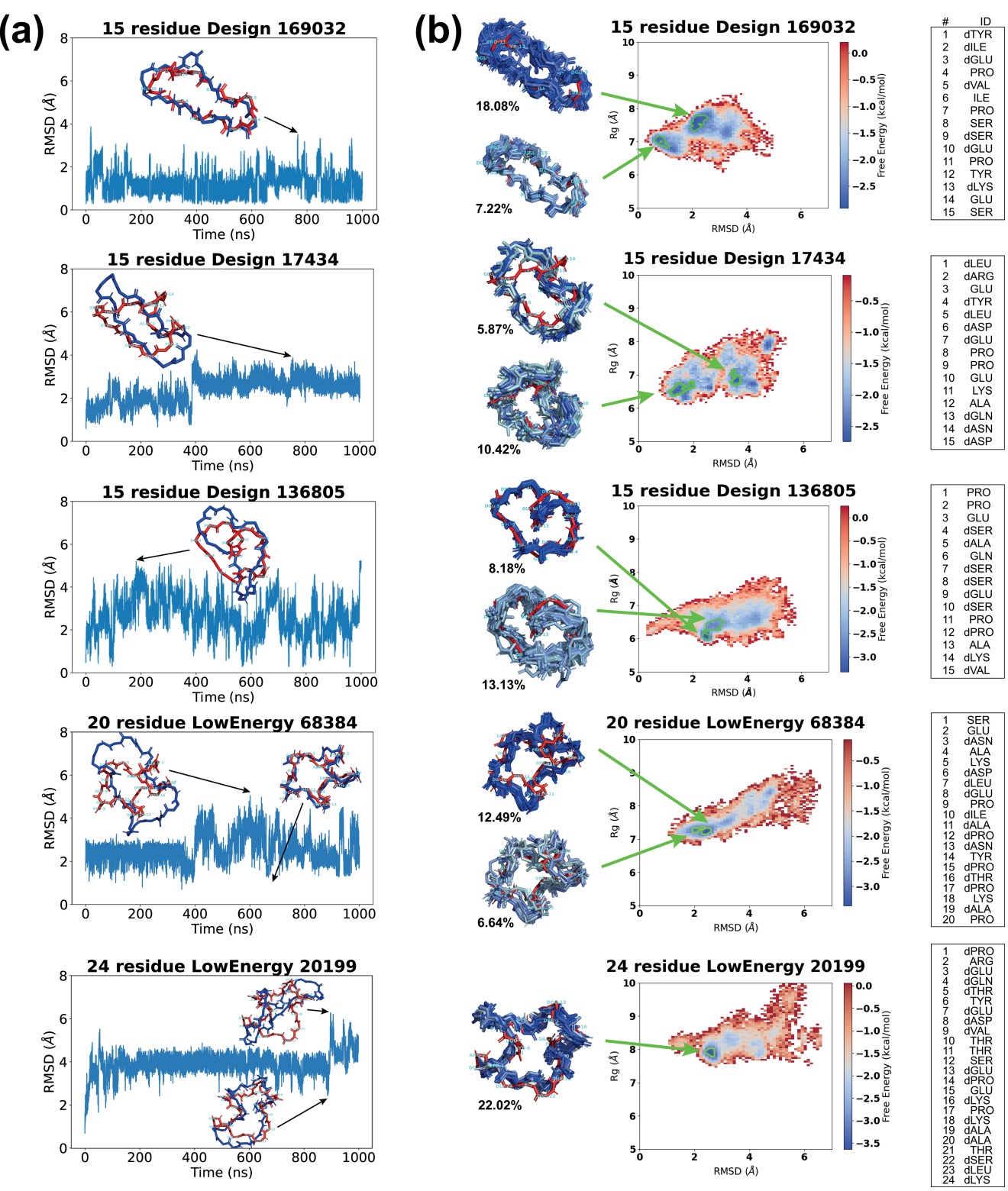

**Fig 8. Molecular dynamics simulation results of top designs.** (a) Backbone $C_\alpha$-atom RMSDs are calculated between the MD trajectory frames and our designed structures. Snapshots are shown for selected time points (designed structures in red, trajectory frames in blue). (b) REMD free energy surfaces. From the lowest free energy basins (marked by green boundaries), representative structures (colored in blue) are extracted from the histogram bins and aligned against our designed structures (red), with the population percentages in the minima labeled aside. The designed sequences are listed on the side.

and assess the thermodynamic stability of these designs, we proceeded with replica exchange molecular dynamics (REMD) simulations.

## Replica Exchange Molecular Dynamics results for stable trajectories

In addition to the nine designs with stable MD trajectories (Fig 8A), we conducted replica exchange molecular dynamics (REMD) simulations on three PDB crystal structures (8-residue PDB 6ucx, 10-resdiue PDB 6uf7, and 12-residue PDB 6uf8) from a prior study [25] as our positive control group. As negative controls, we selected one structure from each size of 8, 10, 12, 15, 20, and 24 residues, randomly permuted its amino acid sequences, and refined its side-chains using Rosetta's FastRelax Cartesian relaxation protocol (S9 Text). These control groups provided a basis for evaluating our designs' REMD outcomes.

We verified the convergence of the REMD simulations by examining physical properties like temperature and radius of gyration. Details and associated plots are provided in S10 Text and S12–S15 Figs. Subsequently, we calculated the average RMSDs of $C_\alpha$-atoms for uncorrelated configurations sampled at 300 K against the original designs, as documented in Table 1.

Both our positive and negative control groups showed noticeable increases in the average RMSDs as the size grew. For smaller macrocycles (around 15 residues), an RMSD around 2 Å indicates a very strong candidate, around 3 Å is good, 4 Å is weak, and greater than 5 Å is poor, with a greater allowance for larger sizes. Consequently, our 15-residue Design 169032, with an RMSD of just 2.21 Å, stood out as a highly promising candidate. Similarly, the other two 15-residue designs 17434 and 136805, plus the 20-residue LowEnergy 68384, and the 24-residue LowEnergy 20199, each with an average RMSD of approximately 3 Å, were also favorable. The remaining high-RMSD designs are considered to have failed the REMD validation (Table 1).

The free energy surfaces (FES) for these five low-RMSD candidates are shown in Fig 8B. As expected, Design 169032 had energy minima in low-RMSD regions. Its lowest energy minimum $E_{min}$ was located at 2.04 Å RMSD. An 18.08% population of the 300 K configurations fell in this energy basin (energy minimum bin and adjacent bins with free energies $\leq 0.9 * E_{min}$). The second lowest energy basin, at an RMSD of 0.71 Å, accounted for 7.22% of the population. We aligned representative structures from these energy basins (one per bin) with the designed structure, and noted minor deviations in the two proline backbone ends (residues 4 and 11).

Design 17434 had more scattered energy basins, one below 2 Å with 10.42% population, and the structures maintained the global conformation of the design. The other basin located at almost 4 Å with 5.87% population, and had more elongated conformations, as was seen in its MD trajectory.

Design 136805 exhibited a "heart-like" configuration, with the backbone curling in the middle. Its free energy surface featured a deep concentrated energy basin at 2.49 Å RMSD, accounting for 8.18% of the population. Structures from this energy basin aligned well with our design, showing deviations mainly in the curl region of the backbone (residues 14 and 15). Nearby, the second-lowest energy minimum was found at 2.71 Å RMSD, presenting a broader basin that encompassed 13.13% of the population. Structures from this basin were more variable, with main differences in the right half of the "heart" (residues 1–6).

LowEnergy 68384 displayed a "star-like" conformation with five arms, and its free energy surface exhibited a diagonal distribution. The two energy minima were situated in the lower left region, featuring compact structures akin to the original design. The structural deviations mainly occurred in two arms (residues 11–13 and residues 19–20+1–4).

**Table 1. Computational validation of top designs using REMD simulations.** For each simulation, the average RMSD is computed for backbone $C_\alpha$ atoms using uncorrelated configurations sampled at temperature state 300 K. As a negative control, average RMSDs are also computed for randomly permuted sequences of each size. By comparing the free energy surfaces of the designed structures with those starting from alternative conformations, the validation results are categorized into computationally validated (the design state is reached in REMD simulations from different starting states), computationally suggestive (the design state is preserved in REMD simulations if started in the design state), and failed (the design state is not preserved in REMD simulations).

| n | Design name | Average RMSD (Å) | Random RMSD (Å) | Computational validation |
|---|---|---|---|---|
| 8 | PDB 6ucx | 0.48 | 2.23 | Computationally suggestive |
| 10 | PDB 6uf7 | 1.31 | 2.92 | Computationally suggestive |
| 12 | PDB 6uf8 | 2.77 | 3.01 | Computationally suggestive |
| **15** | **Design 169032** | **2.21** | 4.49 | Computationally validated |
| **15** | **Design 17434** | **2.92** | | Computationally suggestive |
| **15** | **Design 136805** | **3.24** | | Computationally validated |
| **20** | **LowEnergy 68384** | **3.14** | 5.17 | Computationally validated |
| 20 | LowEnergy 27893 | 4.10 | | Failed |
| 20 | LowEnergy 1665 | 4.12 | | Failed |
| **24** | **LowEnergy 20199** | **3.28** | | Computationally suggestive |
| 24 | LowEnergy 21698 | 4.04 | 6.64 | Failed |
| 24 | LowEnergy 19384 | 5.10 | | Failed |

LowEnergy 20199 had a single deep energy minimum at RMSD 2.5 Å, occupying 22.02% of the population. The structures aligned closely with the design, making this design also promising.

For other designs with high average RMSDs, their free energy surfaces generally showed minima in regions of high RMSD, as depicted in S16 and S17 Figs. Regarding the randomly-permuted sequences included as negative controls, there were notable changes in the FES compared to the designed sequences. In particular, the randomly-permuted version of the 8-residue 6ucx displayed multiple energy minima in its FES, while the FES of the unperturbed sequence showed a single minimum. The randomly-permuted 24-residue 21698 had its energy basins located at much higher RMSD regions, indicating that the original designed backbone was incompatible for the randomized sequence. This suggests the importance of sequence design in maintaining a thermodynamically stable fold.

To further validate that our REMD simulations thoroughly sampled the energy landscapes, we repeated the same protocol for all nine designs and the three PDB control structures, starting from "wrong" backbone conformations. Specifically, from the $P_{Near}$ landscapes generated by our ClusterGen, we picked alternative backbones with low energies but high RMSDs for the designed sequences. If the FES from these "wrong" starting points resemble the original FES, i.e., sampling similar energy minima distributions, it would suggest that our sampling is sufficient and not biased by the starting conformation. A design that has low average RMSD and convergent new FES is said to be *computationally validated* (Table 1).

The comparative results are displayed in S16 and S17 Figs. Notably, for Design 169032, the new FES was almost identical to the original one, further confirming its strong stability and supporting near-exhaustive sampling. Among the other four top candidates, the 15-residue

Design 136805 and the 20-residue LowEnergy 68384 also show a similar new FES compared with the original. Hence, we list these three designs as computationally validated in Table 1.

Interestingly, for the experimentally solved structures 8-mer 6ucx, 10-mer 6uf7, and 12-mer 6uf8, the new FES from the alternative starting structures did not converge well to the original ones, unable to reach the native states. This suggests that designs with non-convergent FES are not necessarily unstable but rather in an ambiguous state where the computational validation could not draw a definitive conclusion. A possible reason might be deep kinetic traps that hinder exploration of the conformation space. We see such non-convergent cases in our 15-residue Design 17434 and 24-residue 20199. We label them as *computationally suggestive* (Table 1).

## Structure predictions for existing macrocycles found in the Protein Data Bank

Finally, we assessed whether ClusterGen could effectively navigate the rugged energy landscapes and identify the lowest energy minima. For 20 existing macrocycles of 7-24 residues without cross-links, whose experimentally-determined structures were previously deposited in the PDB, we applied ClusterGen to generate energy landscapes, using only the amino acid sequences as input (S18 Fig). From each landscape, we then selected the five lowest-energy cluster centers as our structural predictions. The best predictions are displayed in Fig 9, and the remaining in S19 Fig.

For seven PDB-deposited structures of lengths 7–10 residues taken from Hosseinzadeh *et al.*'s 2017 Rosetta design paper [13] (labeled in blue in Fig 9), our lowest-energy predictions achieved RMSDs of between 0.277 and 1.083 Å to the experimentally-determined structure. Ten additional PDB-deposited structures of lengths 8–24 were from Mulligan *et al.*'s Rosetta symmetric design paper [25] (labeled in black in Fig 9). Unlike the original study, which imposed an assumption of internal symmetry during Rosetta structure prediction steps to make $P_{Near}$ landscape generation tractable, we considered all possible conformations without any assumptions, which drastically increased the conformational space and hence the sampling difficulty. Nevertheless, our method still yielded good predictions, having seven out of ten predicted structures within 1.5 Å RMSD. Even for the 24-mer 6uf9, our prediction maintained 1.559 Å RMSD from the crystal structure, and although many of the higher-energy samples returned were asymmetric, the S4 internal symmetry was nicely preserved in the lowest-energy sample. The sole outlier, the 10-mer 6ufu, had an RMSD of 2.331 Å due to the inherent instability of the designed structure, which was confirmed by the presence of two distinct forms in X-ray crystallography experiments.

Besides the Rosetta designed structures, we found three other PDB-deposited macrocycles satisfying our length and no cross-link requirement: 6awm, a 7-mer orbitide (a class of macrocyclic peptides made by plants); 2ns4, a 14-mer peptidomimetic inhibitor featuring short beta sheets and a long loop; and 6dzb, a 16-mer beta sheet structure mimicking an RNA recognition motif. Predictions for these structures were less accurate compared to similarly sized structures. Upon close examination, we found all three were derived from NMR experiments and exhibit a high degree of conformational heterogeneity, compared to other NMR structures for Rosetta designs. Conformational heterogeneity in NMR structures can be due both to flexibility of the peptide itself, or due to limited NMR structural constraints leading to high uncertainty. The observed heterogeneity is particularly evident in the loop regions of 2ns4 and 6dzb, which might be disorganized and floppy. When aligning only the structured beta

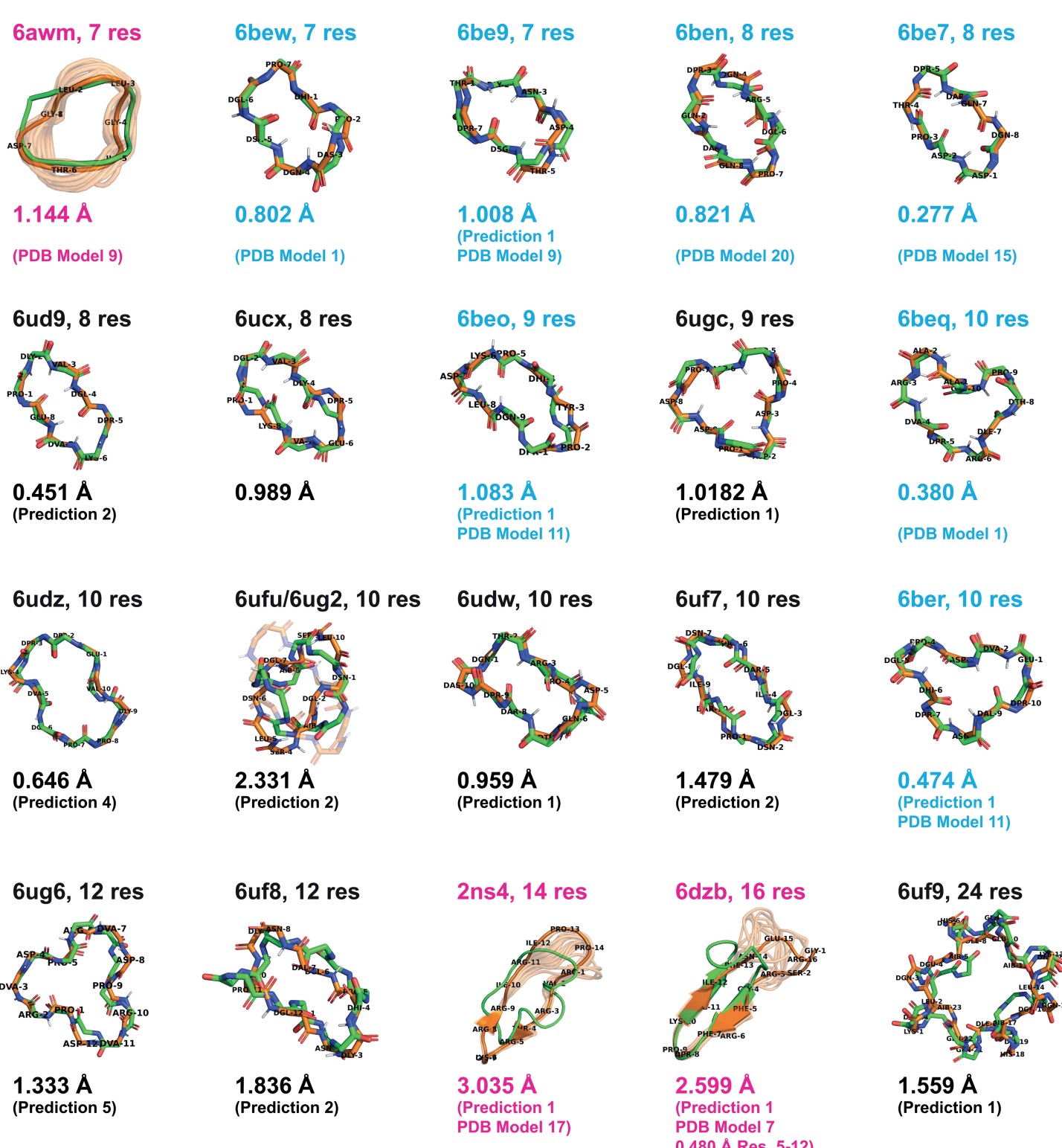

**Fig 9. Structure predictions for macrocycles previously deposited in the PDB.** The best predictions (green) from the low-energy cluster centers are aligned to the PDB structures (orange), with RMSDs shown. PDB structures from the 2017 [13] and 2020 [25] Rosetta design papers are labeled in blue and black, respectively, and all others in pink.

sheet regions (residues 5–12) in 6dzb, our prediction achieved a significantly lower RMSD of 0.480 Å.

Note that as most PDB experimental structures used here contain D-amino acids, and two of the remaining three structures (2ns4 and 6dzb) are in the training dataset for Machine Learning-based methods [17], we could not conduct a meaningful prediction comparison with ML methods.

## Discussion

In this work, we have introduced a pipeline, called CyclicChamp, for cyclic peptide design (Fig 1). Many past works have used a single-shot generalized kinematic closure (GenKIC) algorithm to sample closed macrocycle conformations [5,6,13,21,25,26,40,41]. Unfortunately, the GenKIC technique limits the size of macrocycle for which the conformation space may be extensively explored, either for design or for validation.

By contrast, CyclicChamp performs an iterative search of cyclic backbones with favorable features like strong H-bonds and no steric clashes. Because this produces more viable backbone conformations in less time, CyclicChamp is able to design small macrocycles at lower computational cost, and for the first time, access sizes as large as 24 residues without relying on symmetry or chemical cross-links to limit the accessible conformational space. The basic insight is to transform the cyclic backbone constraint into an error function to allow the use of optimization methods like simulated annealing and genetic algorithms. The optimal simulated annealing parameters were selected from well-spaced random samples of possible parameter value combinations, obtained using combinatorial design [33]. While we have assumed ideal bond angles, bond lengths, and $\omega$ torsion angles to simplify the evaluation of the cyclic error function (Fig 1), generalizations that allow these degrees of freedom to deviate slightly from ideal values during the backbone simulated annealing time steps are possible.

Using these algorithmic ideas, we have generated macrocycles of four sizes. For 7-residue designs, we conducted a comprehensive search of Ramachandran spaces by considering all possible torsion bin center combinations for initial backbone torsion angles. Because the number of torsion bin combinations grows exponentially, for larger designs of 15–24 residues, we randomly selected 100,000 initial combinations. Large pools of backbone candidates with distinct torsion bin strings were generated for 15 and 20 residues (Fig 2).

The sparse clusters found in the 7-residue design were due to the limited torsion bin strings. For 24 residues, while the accessible conformational space exponentially expands with the number of backbone degrees of freedom, only a tiny portion of the conformation space represents backbones with favorable features (e.g. hydrogen bonds) that could be stabilized by suitable choice of sequence. As a result, our backbone simulated annealing algorithm reached its limit. Because the search space grows exponentially, and because solving even simpler discrete version of such problem is *NP*-complete [42], there is no known efficient means of sampling conformations across all sizes, though better heuristic methods like ours can increase the maximum size of peptide that can tractably be sampled and designed. Future studies might experiment with alternative energy models or consider less stringent requirements for cyclic error and repulsive energy when selecting backbone candidates to try to push this limit higher.

After the relaxation and design steps were applied on the clustered backbones, we conducted stability tests on the designs having the lowest energies. For 7-residue designs, both Rosetta's random sampling method and our Ramachandran-stability filtering method were employed to generate energy landscapes. A positive correlation was found to exist between the $P_{Near}$ values computed by the two methods. We observed instances in which the filtering

method explored the low-RMSD regions in the energy landscapes more thoroughly than the random sampling method.

We noted that the optimal energy range for high $P_{Near}$ values did not always correspond to the lowest energy levels (Fig 3). Rosetta's sequence design considers only the desired conformation that one is stabilizing, in order to make the problem tractable; however, the true problem that one wishes to solve is that of maximizing the energy gap between the desired conformation and all alternative conformations. The lack of correlation between the best $P_{Near}$ values and the lowest single-state energies could be that Rosetta has artificial ways of lowering the energy of the designed state, such as adding hydrophobic groups, which tend to stabilize all structures universally instead of uniquely stabilizing the designed structure and maximizing the energy gap between this and alternative states.

The Ramachandran-stability filtering approach can extend to design cyclic peptides with constrained but not fully specified sequences. For instance, to design a stable 7-residue cyclic peptide with alanine residues as the first and fifth amino acids, from the backbone candidate pool sampled by layered simulated annealing, we can identify backbones whose first and fifth residues' torsion angles fall in the Ramachandran space accessible to alanine. This approach allows us to reuse existing pools of backbone candidates.

Starting from 15 residues, Rosetta's method tended to struggle with exploring the low-RMSD regions, and often generated similar round-shaped energy landscapes (Fig 5). To resolve this issue, our ClusterGen algorithm begins with two simulated annealing runs targeting low energy and low RMSD, effectively broadening the RMSD spectrum of the landscape. The subsequent genetic algorithm identifies energy minima through iterations of crossover, mutation, and selection. ClusterGen has successfully differentiated designs of various energy landscape shapes (Fig 5).

In the top 15-residue designs, we observed short alpha helices as depicted in Fig 5. Recurring backbone bendings were induced by $i, i + 3$ H-bonds, which led to more twisted shapes compared to the simple circular backbones seen in 7-residue designs. In the top 20- and 24-residue designs, we saw more diverse secondary structures such as $3_{10}$-helices, $\beta$-bridges, and $\beta$-ladders (Figs 6 and 7). Although complete $\alpha$-helices or $\beta$-sheets are not fully formed, fragments of these structures start appearing, aiding in the stabilization of these mid-sized peptides. Long-range H-bonds also play a crucial role in stabilizing the 20 and 24 residue macrocycles.

Additionally, for 15–24 residues, we see an enrichment of high $P_{Near}$ values in compact structures (Figs 5, 6, and 7). This suggests that simply sorting the designs in ascending order of energies may not be the most effective strategy to identify top designs for stability validation. A more nuanced approach could be to select designs that feature secondary structures or long-range H-bonds and have backbone radii below a specific threshold, and then to sort them by energy.

We have found close backbone matches in our 7-residue designs to the three previously-published, experimentally-solved Rosetta designs (Fig 4). For 15, 20, and 24 residues, we conducted MD and REMD simulations to evaluate the kinetic and thermodynamic stability of our top designs (Fig 8). Specifically, the 15-residue Design 169032 demonstrated exceptional stability, with free energy minima around the 2 Å RMSD region, and the representative structures closely aligning with the design. Two other 15-residue designs 17434 and 136805, plus a 20-residue LowEnergy 68384 and a 24-residue LowEnergy 20199 also managed to preserve their overall shapes throughout the simulation, despite some local conformational movements. Further REMD validations starting from alternative conformations confirmed that Design 169032, 136805, and 68384 had sufficient sampling of FES, as similar low energy basins were found, and we thus considered them as computationally validated designs.

Our ClusterGen's accurate structural predictions of experimentally determined macrocycles (Fig 9), especially for the symmetric 24-mer, demonstrate ClusterGen's ability to overcome the high dimensionality challenge in energy landscape sampling, leading to the successful identification of low-energy native states. This suggests that $P_{Near}$ stability analysis using CluterGen is an effective method for validating designs. Moreover, we provide a new tool for handling general large-sized macrocycle structure prediction, compared to ML-based methods which could only predict sequences with canonical L-amino acids, and Rosetta's *simple_cycpep_predict* which fails to reach low-energy native states beyond 15 residues unless structural constraints are assumed.

To the best of our knowledge, this work represents the first instance of general, unconstrained design of 15-, 20-, and 24-residue mixed chirality macrocycles, without relying on limitation of degrees of freedom through the use of symmetry, disulfides, or other cross-links. The capability to design such large sizes not only enhances the structural diversity of cyclic peptides for future drug search, but also allows larger interaction surfaces for drug binding. Moreover, this opens a door to the design of cyclic-peptide enzymes, which require larger sizes to form active site pockets and may incorporate exotic chemical groups with active-site residues for catalysis.

## Supporting information

**S1 Text. Cyclic error derivation.**
(PDF)

**S2 Text. Backbone energy functions.**
(PDF)

**S3 Text. Layered simulated annealing.**
(PDF)

**S4 Text. Combinatorial design.**
(PDF)

**S5 Text. Torsion angle FastRelax script.**
(PDF)

**S6 Text. Small macrocycle FastDesign script.**
(PDF)

**S7 Text. Large macrocycle FastDesign script.**
(PDF)

**S8 Text. ClusterGen stability analysis.**
(PDF)

**S9 Text. Cartesian coordinate FastRelax script.**
(PDF)

**S10 Text. MD and REMD protocol.**
(PDF)

**S1 Fig. Backbone energy functions.**
(PDF)

**S2 Fig. Extended glycine Ramachandran space sampling.**
(PDF)

**S3 Fig. Ramachandran spaces for L and D amino acids.**
(PDF)

**S4 Fig. ClusterGen crossover and mutation.**
(PDF)

**S5 Fig. ClusterGen computation time breakdown.**
(PDF)

**S6 Fig. CyclicChamp design results.**
(PDF)

**S7 Fig. D-amino acid counts in CyclicChamp designs.**
(PDF)

**S8 Fig. Top 20-residue designs shown in sphere mode.**
(PDF)

**S9 Fig. Top 24-residue designs shown in sphere mode.**
(PDF)

**S10 Fig. Molecular dynamics simulation stable trajectories.**
(PDF)

**S11 Fig. Molecular dynamics simulation unstable trajectories.**
(PDF)

**S12 Fig. REMD temperature plots.**
(PDF)

**S13 Fig. REMD temperature dwell time plots.**
(PDF)

**S14 Fig. REMD convergence check of $R_g$.**
(PDF)

**S15 Fig. REMD convergence check of RMSD.**
(PDF)

**S16 Fig. REMD free energy surfaces with negative controls.**
(PDF)

**S17 Fig. Other REMD free energy surfaces.**
(PDF)

**S18 Fig. Energy landscape predictions of available macrocycle structures deposited in the PDB.**
(PDF)

**S19 Fig. Other structure predictions for available macrocycle structures deposited in the PDB.**
(PDF)

**S1 Table. Backbone simulated annealing parameters.**
(PDF)

**S2 Table. Amino acid sequences of designs having high $P_{Near}$ values.**
(PDF)

## Acknowledgments

We extend our sincerest gratitude to several individuals whose expertise help the successful completion of this work. We would like to thank P. Douglas Renfrew at the Flatiron institute for his suggestions on our data analysis, Bargeen Turzo for the MD package setup, Pilar Cossio, Sonya Hanson, Justin Lindsay, and Miro Astore for their guidance on MD and REMD simulations, and Nick Carriero and Géraud Krawezik for enhancing our codes' parallelism on the Flatiron Institute's HPC. We also would like to thank Shenglong Wang for setting up the NYU HPC. The Flatiron Institute is a division of the Simons Foundation.

## Author contributions

**Conceptualization:** Qiyao Zhu, Vikram Khipple Mulligan, Dennis Shasha.

**Data curation:** Qiyao Zhu.

**Formal analysis:** Qiyao Zhu, Vikram Khipple Mulligan, Dennis Shasha.

**Funding acquisition:** Vikram Khipple Mulligan, Dennis Shasha.

**Investigation:** Qiyao Zhu, Vikram Khipple Mulligan, Dennis Shasha.

**Methodology:** Qiyao Zhu, Vikram Khipple Mulligan, Dennis Shasha.

**Project administration:** Vikram Khipple Mulligan, Dennis Shasha.

**Resources:** Vikram Khipple Mulligan, Dennis Shasha.

**Software:** Qiyao Zhu, Vikram Khipple Mulligan, Dennis Shasha.

**Supervision:** Vikram Khipple Mulligan, Dennis Shasha.

**Validation:** Qiyao Zhu, Vikram Khipple Mulligan, Dennis Shasha.

**Visualization:** Qiyao Zhu, Vikram Khipple Mulligan, Dennis Shasha.

**Writing – original draft:** Qiyao Zhu, Vikram Khipple Mulligan, Dennis Shasha.

**Writing – review & editing:** Qiyao Zhu, Vikram Khipple Mulligan, Dennis Shasha.

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
