## [Decision Letter · Decision Letter 0]

13 Nov 2024

Dear Dr. Shasha,

Thank you very much for submitting your manuscript "Heuristic energy-based cyclic peptide design" for consideration at PLOS Computational Biology.

As with all papers reviewed by the journal, your manuscript was reviewed by members of the editorial board and by several independent reviewers. In light of the reviews (below this email), we would like to invite the resubmission of a significantly-revised version that takes into account the reviewers' comments.

I am aware that this manuscript has been out for review for significantly longer than most manuscripts this journal handles, and I am sorry for this delay. Much of the delay was due to difficulty finding appropriate reviewers. Though we solicited reviews from 27 scientists, we only received one review, and that review lacked practical suggestions (except for the necessity of experimental evidence).

In the absence of sufficient reviewers, the editors discussed this manuscript among ourselves and came to the following conclusion:

Though the method is novel, which should be sufficient for publication here without experiments, the results should be benchmarked compared to known structures. There should be quite a few cyclic peptides of known structure, and for this manuscript to be validated, we would want to see the RMSD of the model vs real for enough of these (perhaps 20?). It seems like Table 1 touches on this, but it is not clear enough nor does it have enough examples.

I understand that you are already using MD for validation. I think for MD to be a validation tool, I'd like to see simulations with wrong cyclic peptide structures. And I'd like to see that the simulations are different enough from the right structure to be able to tell right from wrong (For 20 cases).

Finally, though I understand AF/ESM/RoseTTAFold don't yet handle D-amino acids, if any of your designs are all L, could you please validate the structure using your structure prediction method of choice?

We understand that the changes we suggest might be too extensive to handle within the stipulated 90 days, so we request a major revision. If you find that the requests are too extensive to handle within 90 days, please get in touch with the editorial board and tell them that you either will submit the paper elsewhere or that you will resubmit it here when you are ready.

We cannot make any decision about publication until we have seen the revised manuscript and your response to the reviewers' comments. Your revised manuscript is also likely to be sent to more reviewers for further evaluation.

Sincerely,

Joanna Slusky, Ph.D.

Academic Editor

PLOS Computational Biology

Nir Ben-Tal

Section Editor

PLOS Computational Biology

Reviewer's Responses to Questions

**Comments to the Authors:**

Reviewer #1: The paper titled "Heuristic Energy-Based Cyclic Peptide Design" introduces a new computational pipeline, CyclicChamp, designed to efficiently generate de novo cyclic peptides. This innovative approach marks a significant improvement over previous random sampling methods by employing simulated annealing, making it faster and more capable of handling larger macrocycles of up to 24 amino acids. The research combines molecular dynamics simulations to validate the kinetic and thermodynamic stability of the peptides, leading to promising candidates for future experimental evaluation.

Strengths are:

1.Innovative Methodology

2.Relevant Applications

3. Accurate computational validation

4. Efficiency and scalability

However, the manuscript completelly lacks any experimental validation.

Therefore, I believe the manuscript should be accepted only after - some - experimetal validation is provided.

**Have the authors made all data and (if applicable) computational code underlying the findings in their manuscript fully available?**

Reviewer #1: Yes

PLOS authors have the option to publish the peer review history of their article (what does this mean?). If published, this will include your full peer review and any attached files.

Reviewer #1: No
---

## [Editor Report · Decision Letter 1]

27 Feb 2025

Dear Dr. Shasha,

We are pleased to inform you that your manuscript 'Heuristic energy-based cyclic peptide design' has been provisionally accepted for publication in PLOS Computational Biology.

Best regards,

Joanna Slusky, Ph.D.

Academic Editor

PLOS Computational Biology

Nir Ben-Tal

Section Editor

PLOS Computational Biology

---

## [Editor Report · Acceptance letter]

PCOMPBIOL-D-24-01108R1

Heuristic energy-based cyclic peptide design

Dear Dr Shasha,

I am pleased to inform you that your manuscript has been formally accepted for publication in PLOS Computational Biology. Your manuscript is now with our production department and you will be notified of the publication date in due course.

With kind regards,

Zsofia Freund
